# SIRT7 promotes genome integrity and modulates non-homologous end joining DNA repair

Berta N Vazquez[1], Joshua K Thackray[1], Nicolas G Simonet[2], Noriko Kane-Goldsmith[1], Paloma Martinez-Redondo[2], Trang Nguyen[1], Samuel Bunting[3], Alejandro Vaquero[2], Jay A Tischfield[1] & Lourdes Serrano[1,*]

## Abstract

Sirtuins, a family of protein deacetylases, promote cellular homeostasis by mediating communication between cells and environment. The enzymatic activity of the mammalian sirtuin SIRT7 targets acetylated lysine in the N-terminal tail of histone H3 (H3K18Ac), thus modulating chromatin structure and transcriptional competency. SIRT7 deletion is associated with reduced lifespan in mice through unknown mechanisms. Here, we show that SirT7-knockout mice suffer from partial embryonic lethality and a progeroid-like phenotype. Consistently, SIRT7-deficient cells display increased replication stress and impaired DNA repair. SIRT7 is recruited in a PARP1-dependent manner to sites of DNA damage, where it modulates H3K18Ac levels. H3K18Ac in turn affects recruitment of the damage response factor 53BP1 to DNA double-strand breaks (DSBs), thereby influencing the efficiency of non-homologous end joining (NHEJ). These results reveal a direct role for SIRT7 in DSB repair and establish a functional link between SIRT7-mediated H3K18 deacetylation and the maintenance of genome integrity.

**Keywords** DNA damage; histone acetylation; non-homologous end joining; PARP1; SIRT7
**Subject Categories** Chromatin, Epigenetics, Genomics & Functional Genomics; DNA Replication, Repair & Recombination; Post-translational Modifications, Proteolysis & Proteomics
**The EMBO Journal (2016) 35:** 1488–1503

See also: **S Paredes & KF Chua** (July 2016)

## Introduction

Sirtuins are NAD$^+$-dependent protein deacetylases and, in some cases, NAD$^+$-dependent ADP ribosyltransferases. Mammals have seven sirtuin family members, which are denoted SIRT1 to 7. They are involved in sensing and responding to different types of cellular stressors, including fasting, genotoxic, and oxidative stress (Vaquero & Reinberg, 2009). A key function of sirtuins is the regulation and maintenance of genome stability under stress. As this regulation fails, genome integrity can diminish, resulting in devastating consequences on cellular fitness, cumulatively leading to organismal aging. Indeed, numerous studies from yeast to mice support a role for sirtuins in the amelioration of human aging-related pathologies (Guarente, 2013), and its deletion is associated with genome instability and compromised organismal viability (Cheng *et al*, 2003; McBurney *et al*, 2003; Mostoslavsky *et al*, 2006; Wang *et al*, 2008a; Serrano *et al*, 2013).

Deficiency of SIRT7 has been linked to several pathologies including cardiac hypertrophy (Vakhrusheva *et al*, 2008b; Ryu *et al*, 2014), hepatic steatosis (Shin *et al*, 2013; Ryu *et al*, 2014), and deafness (Ryu *et al*, 2014). SIRT7 has been functionally linked to transcriptional regulation. SIRT7 is detected at promoters and coding regions of ribosomal genes, where it positively controls ribosome production through direct interaction with the PolI machinery (Ford *et al*, 2006; Grob *et al*, 2009; Chen *et al*, 2013). Conversely, SIRT7 negatively regulates the transcription of genes outside of the rDNA repeats via histone H3K18 deacetylation (Barber *et al*, 2012). Recent evidence indicates that SIRT7 may have the capacity to act as an oncogene as its expression is elevated in several human cancers (Roth & Chen, 2014). The oncogenic potential of SIRT7 has been attributed to the transcriptional regulation of a specific set of genes through direct interaction with the EKL4 transcription factor. However, whether or not the oncogenic potential of SIRT7 is due to the promotion of genome instability remains heretofore unexplored.

Here, we investigate the role of SIRT7 in the maintenance of genome integrity by characterizing *SirT7$^{-/-}$* SIRT7-knockout (KO) mice. This mouse model has been shown previously to develop fatty liver due to transcriptional-related ER stress (Shin *et al*, 2013). Our results show that these mice also suffer from reduced embryonic viability and the development of a progeroid-like phenotype in mice that survive to adulthood. Moreover, SIRT7-deficient cells present replication stress and impaired DDR. In addition, we show that SIRT7 deletion correlates with increased H3K18 acetylation at DNA damage sites and impaired NHEJ repair, revealing a novel molecular

1 Department of Genetics, Human Genetics Institute of New Jersey, Rutgers University, Piscataway, NJ, USA
2 Chromatin Biology Laboratory, Cancer Epigenetics and Biology Program (PEBC), Bellvitge Biomedical Research Institute (IDIBELL), Barcelona, Spain
3 Department of Molecular Biology and Biochemistry, Rutgers University, Piscataway, NJ, USA
*Corresponding author. Tel: +1 848 445 9577; E-mail: serrano@biology.rutgers.edu

mechanism for H3K18Ac in the maintenance of genome integrity, in addition to transcriptional regulation. This report provides new evidence for the role of sirtuins in aging, both by acting as functional mediators between environmental and metabolic factors, and in the regulation of genome stability.

# Results

### Perinatal lethality and accelerated aging phenotype in SIRT7-deficient mice

We first determined the consequences of SIRT7 loss (Appendix Fig S1A–C) on mouse embryonic development and lifespan. Remarkably, after intercrossing $SirT7^{+/-}$ parental mice, we observed that $SirT7^{-/-}$ pups were born at sub-Mendelian ratios in 129S1/SvImJ mice (9%, down from the predicted 25%, P-value < 0.001 by $\chi^2$ test, Fig 1A)

and in C57/BL6 $SirT7^{-/-}$ mice (4%, down from the predicted 25%, Appendix Fig S1D), indicative of a defect in embryogenesis, which was still observed at gestational day 14.5 but to a substantially reduced extent (20%, down from predicted 25%; Fig 1A). The weight at birth was lower in $SirT7^{-/-}$ mice relative to WT (5.06 ± 0.1 and 6.8 ± 0.04 g, respectively; Fig 1B and C, and Appendix Fig S1E). Over the 26-week period examined, the rate of growth was similar in the two groups (slope); however, they maintained their weight differences over this time frame as indicated by the graph plateaus. Importantly, more than 20% of $SirT7^{-/-}$ mice died within the first month of life, and the remaining KOs died sooner (12–20 months period) than their WT littermates (Fig 1D). Taken together, these results show that SIRT7 depletion is detrimental to lifespan, suggesting a novel role for SIRT7 protein in perinatal development.

Adult $SirT7^{-/-}$ mice showed phenotypic and molecular signs of accelerated aging such as premature (6 months) and pronounced curvature of the spine, kyphosis (100% penetrance; Fig 2A), and

**A**

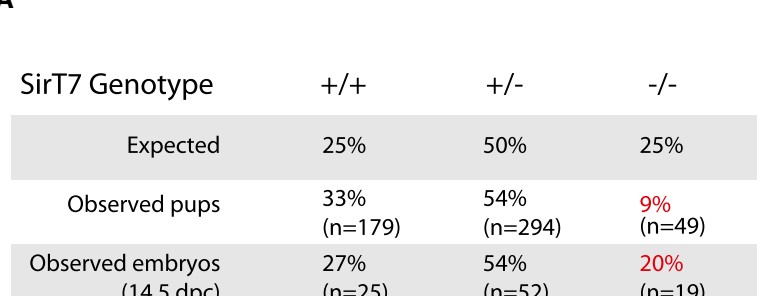

| SirT7 Genotype | +/+ | +/- | -/- |
|---|---|---|---|
| Expected | 25% | 50% | 25% |
| Observed pups | 33% (n=179) | 54% (n=294) | 9% (n=49) |
| Observed embryos (14.5 dpc) | 27% (n=25) | 54% (n=52) | 20% (n=19) |

**B**

WT    SirT7^{-/-}

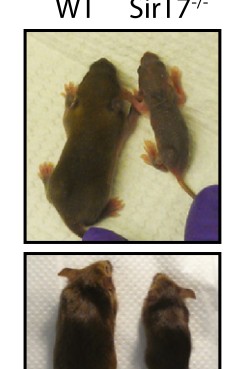

**C**

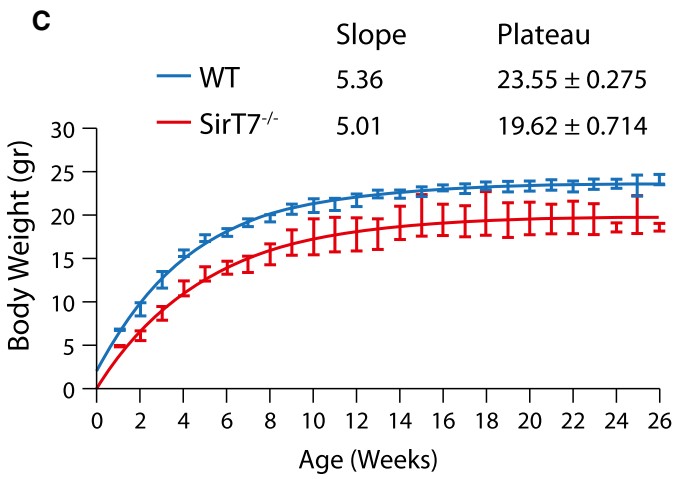

|  | Slope | Plateau |
|---|---|---|
| WT | 5.36 | 23.55 ± 0.275 |
| SirT7^{-/-} | 5.01 | 19.62 ± 0.714 |

**D**

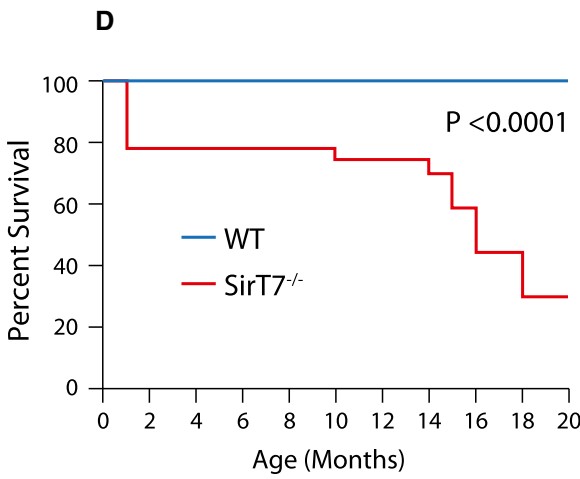

P <0.0001

Figure 1. $SirT7^{-/-}$ mice have increased perinatal lethality and reduced life span.
A   Mendelian ratios from $SirT7^{+/-}$ cross (n = 522 pups; n = 96, 14.5 embryos; P-value < 0.001 by $\chi^2$).
B   WT and $SirT7^{-/-}$ mice at 10 days (top) and 2 months (bottom) of age.
C   Weight distribution of WT and $SirT7^{-/-}$ mice (n = 3–10 female mice per time point and genotype; P-value = 0.003 by unpaired t-test; mean ± SD).
D   Kaplan–Meier survival curves (n = 170 WT and n = 58 $SirT7^{-/-}$ mice; log-rank test P-value < 0.0001).

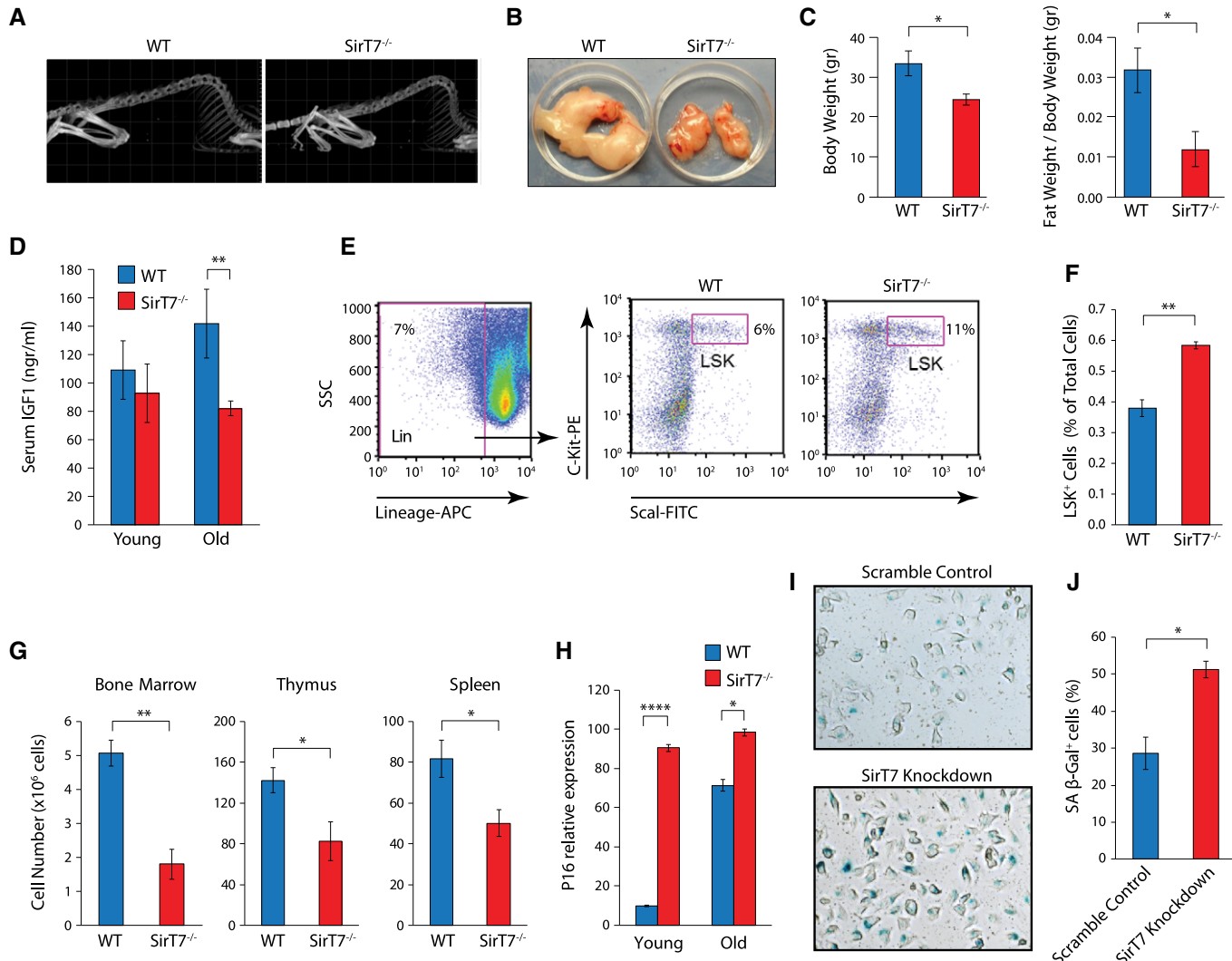

**Figure 2. Accelerated aging phenotype in *SirT7*$^{-/-}$ mice.**

A    Representative 3D reconstructed CT scans showing increased kyphosis in 16-month-old *SirT7*$^{-/-}$ mice compared with WT.

B    Representative gonadal fat pads from WT and *SirT7*$^{-/-}$ 16-month-old mice.

C    Quantitation of body weight (left) and gonadal fat pad mass normalized to total body weight (right) (mean ± SEM; four samples per genotype).

D    IGF-1 protein levels in serum measured by ELISA (mean ± SEM; 3–8 mice per genotype and age-group).

E, F    Dot plots of Lin$^-$Sca1$^+$cKit$^+$ (LSK) cells (middle and right) from WT and *SirT7*$^{-/-}$ bone marrow cells. Cells were gated for negative staining of lineage markers B220, CD3, CD11b, CD19, Gr-1, and Ter-119 (left), and analyzed for Sca1 and cKit expression (middle and right). (F) Quantitation of (E) (mean ± SEM; 4 mice per genotype).

G    Bone marrow, thymus, and spleen cell number in young WT and *SirT7*$^{-/-}$ mice (mean ± SEM; 3–5 mice per genotype).

H    mRNA expression of *p16* gene normalized to GAPDH measured by RT–PCR from young and old WT and *SirT7*$^{-/-}$ fibroblasts (mean ± SEM; three samples per genotype).

I, J    Senescence-associated β-galactosidase staining (I, blue) in HT1080 cells transfected with scramble control or SirT7 knockdown and grown for 7 days in selection media. (J) Quantitation of the number of senescent cells shown in (I) (mean ± SEM; three independent cell lines per genotype).

Data information: *$P < 0.05$; **$P < 0.01$; ***$P < 0.001$; ****$P < 0.0001$ by ANOVA single factor.

decreased gonadal fat pad content (Fig 2B and C). Nevertheless, *SirT7*$^{-/-}$ mice presented increased hepatic lipid content (Fig EV1A), as previously documented (Shin *et al*, 2013). In addition, 14 month-old *SirT7*$^{-/-}$ mice had reduced IGF-1 levels in plasma compared with WT littermates (Fig 2D), as has been observed in different human and mouse progeroid syndromes (Niedernhofer *et al*, 2006; Murga *et al*, 2009).

Aging is also associated with stem cell dysfunction (Sharpless & DePinho, 2007). In the bone marrow hematopoietic stem cell LSK

(Lineage$^-$, Sca1$^+$, cKit$^+$) population, aging is associated with increased cellularity, which is believed to compensate for the lack of regenerative potential as these cells age (Sudo *et al*, 2000). The fraction of LSK-positive cells was increased in young (4-month-old) *SirT7*$^{-/-}$ bone marrow compared with WT littermates (Fig 2E and F). However, the capacity of tissue regeneration of this cell population was reduced as measured by competitive bone marrow transplantation experiments (Fig EV1B and C). WT or *SirT7*$^{-/-}$ bone marrow-derived cells (CD45.2) were mixed with WT cells (CD45.1) in a

1:1 ratio and were used to reconstitute lethally irradiated mice (Fig EV1B). We found that SIRT7-deficient cells had a reduced (~50%) capacity to repopulate the lymphoid compartment compared with WT bone marrow cells (Fig EV1C). Moreover, analysis of primary and peripheral lymphoid organs revealed leukopenia (Fig 2G), as has been reported in other progeroid mouse models including $SirT6^{-/-}$ mice (Mostoslavsky *et al*, 2006; Murga *et al*, 2009).

In addition, we measured *p16INK4* mRNA levels in splenocytes and ear fibroblasts at 4 and 14 months (Fig 2H, fibroblast data not shown). *p16INK4* is a cell cycle regulator that limits the regenerative capacity of many tissues (Krishnamurthy *et al*, 2004). *p16INK4* transcript levels increased as the mice aged and were higher in $SirT7^{-/-}$ mice compared with WT. This difference was markedly greater in younger animals, suggesting premature cellular senescence in $SirT7^{-/-}$ mice.

### SIRT7 participates in the maintenance of genome integrity

Despite the fact that we observed signs of premature senescence in $SirT7^{-/-}$-derived cells (Fig 2H), we did not observe major differences in the cell growth (Appendix Fig S2A) and cell cycle profiles (Appendix Fig S2B) between $SirT7^{-/-}$ and WT MEFs, in agreement with previous reports (Vakhrusheva *et al*, 2008a). Remarkably, at later cell passages, we found a mild G2/M arrest and a twofold increase in both polyploid and apoptotic cells (sub-G1) in $SirT7^{-/-}$ MEFs compared with WT (Fig 3A and B). Consistently, $SirT7^{-/-}$ thymocytes had an increased susceptibility to apoptosis after exposure to different doses of X-ray irradiation (IR) compared with WT (Fig 3C), indicative of sustained DNA damage after IR. We further analyzed cell viability by performing colony formation assays after different X-ray doses in HT1080 cells that had been transfected with either scramble (Scr) or $SirT7$ shRNA. SIRT7-depleted cells formed fewer colonies compared with control cells, once more indicating that SIRT7-deficient cells are more sensitive to externally induced DNA damage (Figs 3D and EV3D). In agreement with the increase in the senescence marker *p16INK4*, knockdown (KD) of SIRT7 in HT1080 cells resulted in increased number of cells positive for the senescence-associated β-galactosidase marker (Fig 2I and J, and Appendix Fig S6A).

We proceeded to examine genome stability in the absence of SIRT7 at the organismal level. We investigated mutation burden in $SirT7^{-/-}$ mice by performing an *in vivo* mutagenesis assay using the adenine phosphoribosyltransferase (APRT) mouse model, a unique system designed to select for loss of heterozygosity *in vivo* (Shao *et al*, 1999) (Fig 3E). Mutant frequency was increased sixfold in $SirT7^{-/-}$ mice compared with WT (Fig 3F), indicating that SIRT7 deficiency increases functional loss of heterozygosity *in vivo*. Overall, these results reveal a functional link of SIRT7 to the maintenance of genome integrity, which might be a consequence of the accumulation of genomic damage. Indeed, we observed increased levels of DNA damage in SIRT7-deficient cells by comet assays (Fig 4A and Appendix Fig S3A). We next questioned whether the DNA damage might be due to impaired DDR, by monitoring the activity of ataxia telangiectasia mutated (ATM), an apical player of DDR (Shiloh & Ziv, 2013), and its downstream effector KRAB-associated protein 1 (KAP-1) (Ziv *et al*, 2006) (Fig 4B). The phosphorylation levels of ATM and target proteins increased upon induced DNA damage, and the induction was more elevated in $SirT7^{-/-}$ cells.

Overall, these results indicate proper DDR in the absence of SIRT7, but confirm the increased DNA damage observed in $SirT7^{-/-}$ cells. This observation was further examined by measuring the double-strand break (DSB) marker and ATM-targeted protein, γH2AX (Lobrich *et al*, 2010), by immunofluorescence (IF). We performed a quantitative analysis of the spatial and cell cycle distribution of γH2AX foci in exponentially growing primary mouse fibroblasts (Fig 4C and D). Sites of active DNA replication during S-phase progression were identified by the incorporation of EdU, and G1 and G2 cells were segmented according to their nuclear volume based on DAPI staining of 3D-reconstructed individual nuclei (Serrano *et al*, 2011) (Appendix Fig S4A). We found a higher number of γH2AX foci throughout the cell cycle in SIRT7-deficient than in WT cells (Fig 4C and D, NoIR). The temporal dynamics of repair, as measured by the time required for a reduction in foci to non-irradiated levels, was similar in the two cell types (Fig 4D, from NoIR to IR-8 h). Eight hours after IR, SIRT7-depleted cells showed increased number of γH2AX foci per nucleus compared with WT, to a similar extent as was observed before IR. We next explored whether DNA damage was associated with a specific type of chromatin by analyzing the nuclear localization of γH2AX foci. We enumerated the number of γH2AX foci overlapping or within the periphery of pericentric heterochromatin (HC) assessed by DAPI staining (Fig 4E and F, G2 is shown; Appendix Fig S4B). Euchromatic numbers were estimated by subtracting the heterochromatic from the total foci number. In agreement with previous reports (Goodarzi *et al*, 2010), euchromatin- and heterochromatin-associated DSBs were repaired with fast and slow kinetics, respectively (Fig 4E). The number of DSBs associated with pericentric heterochromatin regions was similar between WT and $SirT7^{-/-}$ cells before and at all time points after IR.

We next questioned whether SIRT7 deletion leads to replication-associated defects. To test this possibility, WT and $SirT7^{-/-}$ MEFs were treated with hydroxyurea (HU) to promote replication fork stalling and DSB formation during S-phase. At 24 h after HU treatment, we observed in both cell types that HU led to S-phase arrest. However, this was more prominent in SIRT7-deficient cells compared with WT cells (Fig 4G, HU), suggesting that SIRT7-deficient cells are more prone to replication-associated stress. To directly examine this possibility, DNA fiber labeling analysis was used to assess DNA replication fork progression in WT and $SirT7^{-/-}$ primary MEFs. Loss of SIRT7 resulted in a significant reduction in DNA replication fork velocity in $SirT7^{-/-}$ primary MEFs as compared with WT, which became exacerbated as cells were kept in culture (Fig 4H and I). Reduced replication rate often correlates with increased activation of dormant origins of replication, which is believe to be an attempt to rescue global replication rate after forks collapsed (Zhong *et al*, 2013). Consistently, we observed a remarkable increase in the presence of stalled replication forks and in the firing of new origins of replication upon replication block with HU in $SirT7^{-/-}$ primary MEFs as compared with WT (Fig EV2A–C).

Overall, our results show that SIRT7 deficiency leads to replication stress, which has an important impact on genome stability and could contribute to explain the observed progeroid phenotype.

### Impaired double-strand break repair in the absence of SIRT7

Sirtuins participate in the DNA damage response (DDR) by regulating cell cycle progression and DNA repair, particularly

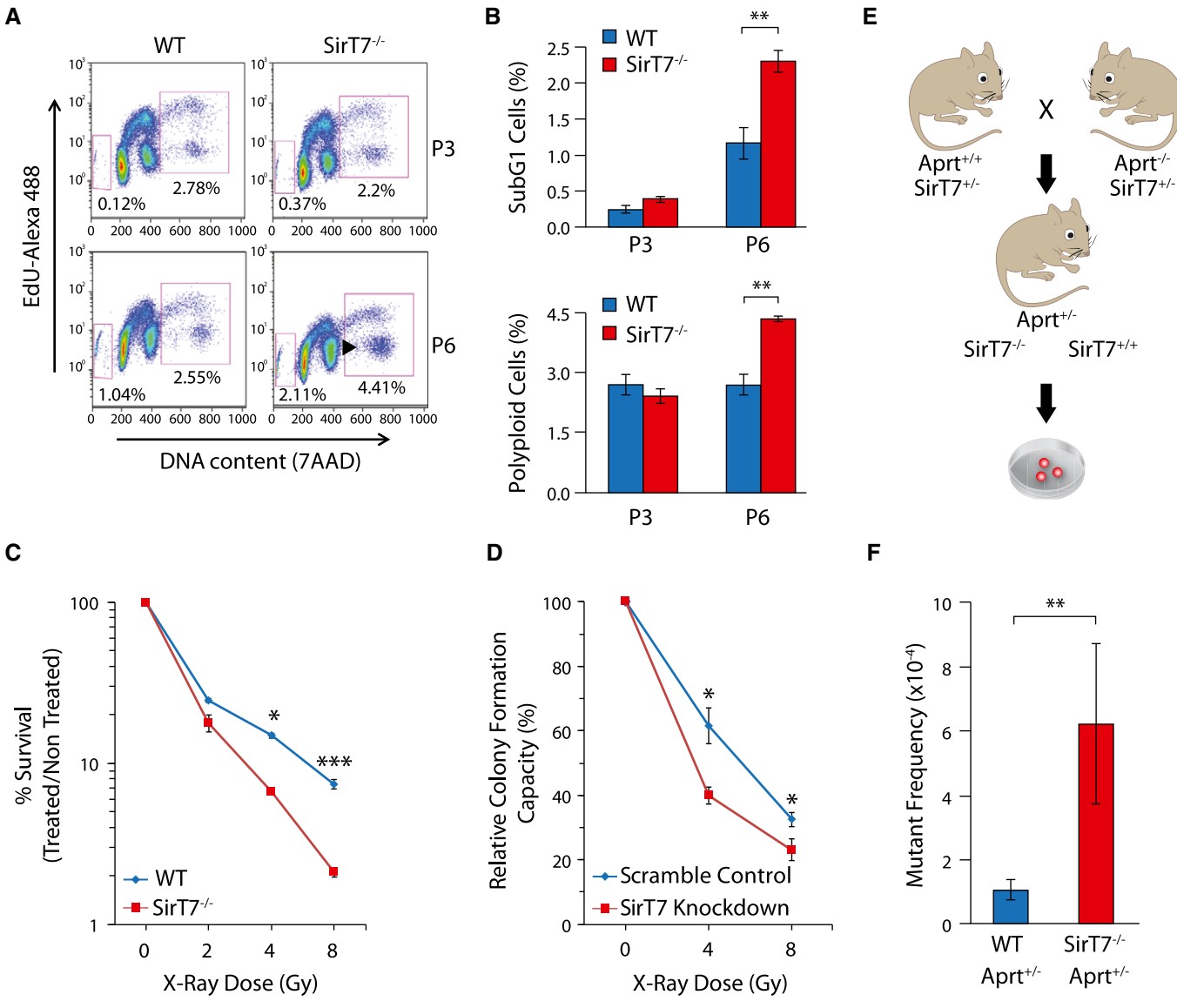

**Figure 3. Increased genome instability in SirT7⁻/⁻ mice.**

A, B    Dot plots of FACS cell cycle analyses of WT and *SirT7⁻/⁻* MEFs in passages 3 (P3) and 6 (P6) using EdU incorporation and 7AAD (A). Percentages of cells in sub-G1 (apoptotic cells, left square) and cells with DNA content above 4N (polyploid, right square). (B) Quantitation of experiment shown in (A) (mean ± SEM; three samples per genotype).

C    Survival curve for WT and *SirT7⁻/⁻* thymocytes after X-ray irradiation (IR) at the indicated doses. Cell death was quantified by FACS using Annexin V and 7AAD staining 18 h postinsult (mean ± SEM; three samples per genotype from one of two independent experiments).

D    Clonogenic assays in HT1080 cells transfected with scramble control or SirT7 knockdown and irradiated at the indicated X-ray doses, then plated at low density. After 9 days, colonies were stained with crystal violet and counted (mean ± SEM; three independent cell lines per genotype).

E    Schematic describing the mouse APRT loss of heterozygosity (LOH) assays. *Aprt⁺/⁻* mice were used to measure the *in vivo* somatic mutation frequency in WT and *SirT7⁻/⁻* mice. Cells fully deficient for APRT are selected in culture by 2,6-diaminopurine (DAP), an adenine analog that is converted to a toxic product by APRT enzymatic activity. Mutant frequency is proportional to the number of DAP-resistant (DAPʳ) colonies.

F    Quantitation of experiment shown in E (mean ± SEM; 9 WT and 3 *SirT7⁻/⁻* samples).

Data information: *$P < 0.05$; **$P < 0.01$; ***$P < 0.001$ by ANOVA single factor.

non-homologous end joining (NHEJ) and homologous recombination (HR), which repair DNA DSBs (Jeong *et al*, 2007; Yuan *et al*, 2007; Li *et al*, 2008; McCord *et al*, 2009; Kaidi *et al*, 2010; Serrano *et al*, 2013; Toiber *et al*, 2013). Similarly, SIRT7 may participate in the maintenance of genome integrity by modulating DSB repair. To test this, we analyzed 53BP1 chromatin focus formation by IF

(Fig 5A). 53BP1 is a repair protein that promotes NHEJ by protecting DNA from end resection (Panier & Boulton, 2014). We observed a remarkably reduced number of 53BP1 foci per nucleus in the absence of SIRT7 before (Fig EV3A) and after inducing DNA damage by IR (Figs 5B and EV3B). Total 53BP1 protein levels were similar between WT and *SirT7⁻/⁻* cells (Appendix Fig S3B),

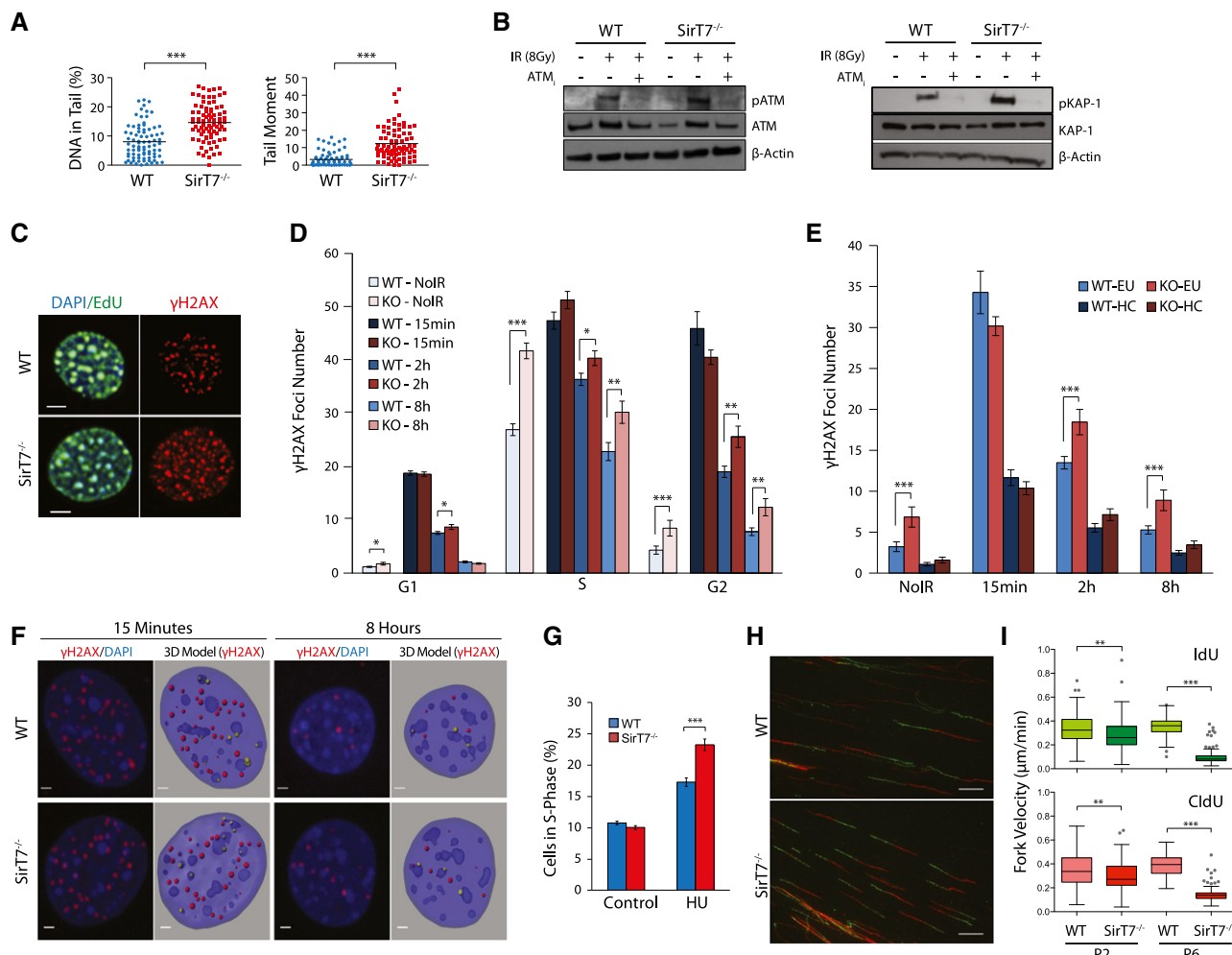

**Figure 4. SIRT7 protects cells from endogenous and induced DNA damage.**

A    Quantitation of neutral comet assays, using passage 3 WT and *SirT7*$^{-/-}$ primary MEFs, showing (left) the amount (%) of DNA in the tail and (right) the tail moment (see Fig S3A for representative images).

B    Western blot showing ATM and KAP1 phosphorylation and total protein levels in WT and *SirT7*$^{-/-}$ primary fibroblasts after IR (8 Gy). ATM inhibitor (ATM$_i$) KU-55933 was added 30 min prior to irradiation where indicated. One representative blot from four independent experiments is shown.

C–F    IF analysis of WT and *SirT7*$^{-/-}$ primary fibroblasts showing γH2AX dynamics after DNA damage induction. Cells were untreated (NoIR) or treated with 1 Gy of X-rays (IR) and fixed at the indicated times postinsult. Cells were pulsed with EdU (green) 30 min prior to fixation, then stained for γH2AX (red), and counterstained with DAPI (blue) (*n* > 30 cells per group/mice; 3 mice per genotype). (C) Representative images of untreated S-phase (NoIR, EdU positive) WT and *SirT7*$^{-/-}$ nuclei (scale bar 5 μm). (D) Quantitation of the number of γH2AX foci per nucleus at the indicated time points post-IR (1 Gy) and indicated cell cycle phases (mean ± SEM from three independent experiments). (E) Quantitation of experiment described in (C) showing mean number of γH2AX foci per nucleus in euchromatic and heterochromatic regions at the indicated time points before and after IR in G2 (mean ± SEM from three independent experiments). Total γH2AX foci and γH2AX foci overlapping with or at the periphery of heterochromatic regions were enumerated. Nuclei and pericentric heterochromatin (chromocenters) were segmented by DAPI staining. Euchromatic numbers were estimated by subtracting the heterochromatic number of foci from the total foci number. (F) Representative images of WT and *SirT7*$^{-/-}$ primary fibroblasts in G2-phase showing γH2AX (red) and DAPI (blue) at the indicated period of time after IR. (Right) 3D rendering of the IF segmentation depicting nuclei (pale blue), chromocenters (darker blue), and γH2AX (yellow denotes foci associated with pericentric heterochromatin, otherwise foci are red). Scale bar 2 μm.

G    FACS quantitation of WT and *SirT7*$^{-/-}$ MEF cells in S-phase after insult with 10 mM hydroxyurea (HU) for 24 h. Cells were fixed and stained with 7AAD and cell cycle was monitored by FACS (mean ± SEM; five samples per genotype).

H, I    DNA fiber labeling analysis was used to assess DNA replication fork progression in passage 3 and passage 6 primary WT and *SirT7*$^{-/-}$ MEFs. (H) Representative images from cells labeled for 20 min with IdU (green) followed by 20 min of CldU (red). (I) Quantitation of fork velocity (fiber length/labeling time; mean ± SEM; three samples per genotype per condition). Scale bar 10 μm.

Data information: *$P$ < 0.05; **$P$ < 0.01; ***$P$ < 0.001 by ANOVA single factor.
Source data are available online for this figure.

suggesting that SIRT7 depletion specifically impacts 53BP1 binding to chromatin. Noticeably, the mean volume of 53BP1 foci per nucleus was reduced in cells from *SirT7*$^{-/-}$ mice compared with WT (Fig 5C). Although 53BP1 facilitation of NHEJ, to the detriment of HR, is plausibly being exerted throughout the cell cycle, 53BP1 also participates in HR-mediated repair of heterochromatin, which is

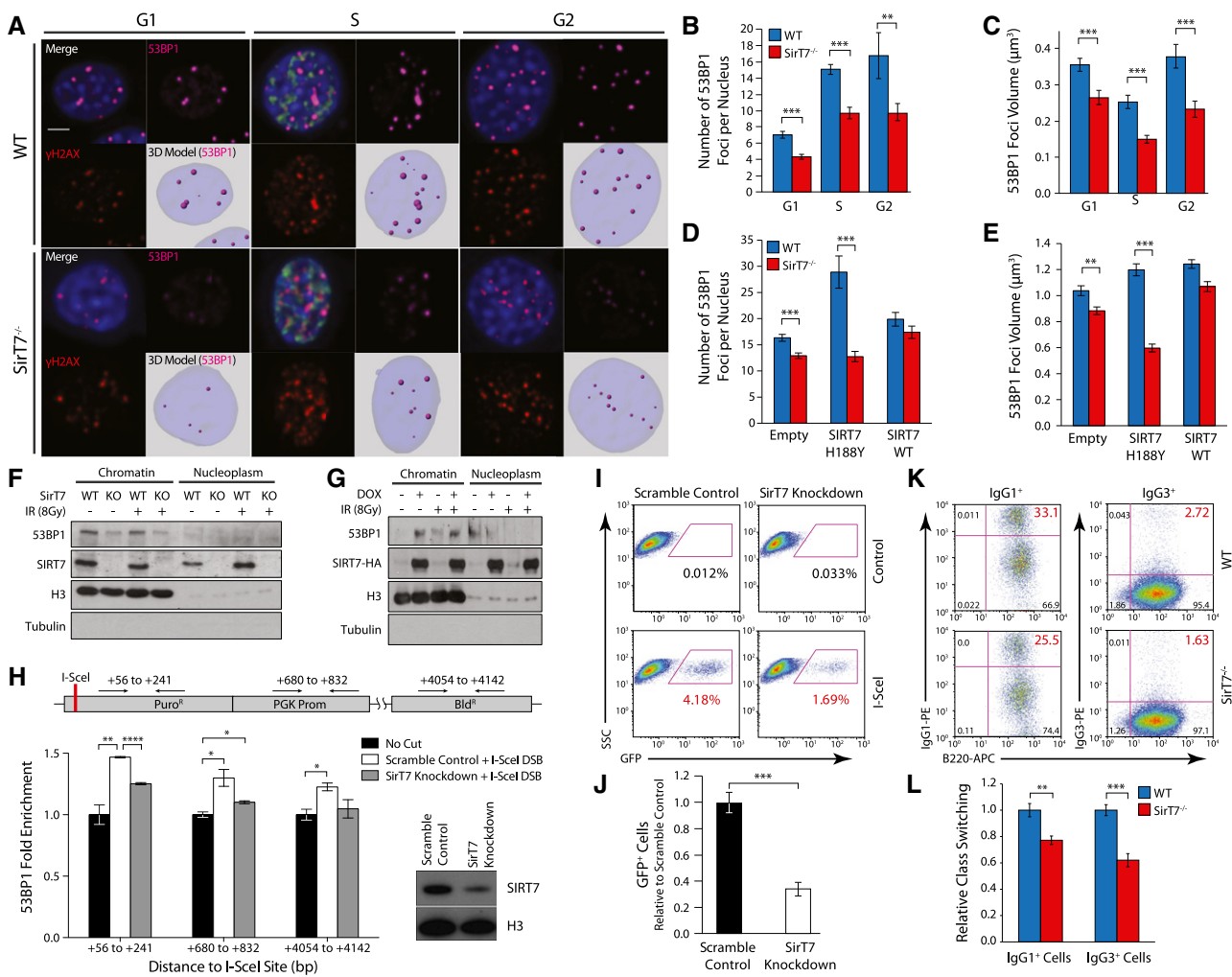

**Figure 5. Impaired NHEJ DNA repair pathway in *SirT7*<sup>−/−</sup> cells.**

A–E   IF analysis of WT and *SirT7*<sup>−/−</sup> primary fibroblasts after IR (1 Gy) and 1-h chase. Cells were pulsed with EdU 30 min prior to fixation, stained with antibodies against γH2AX and 53BP1, and then counterstained with DAPI (scale bar 5 μm). (A) Representative images from each cell cycle stage showing merge (top left) including EdU (green), 53BP1 (top right, magenta), γH2AX (bottom left, red), and 3D rendering of reconstructed Z-stacks with 53BP1 foci modeled as spheres (bottom right; pale blue: nucleus; magenta spheres: 53BP1 foci). (B) Quantitation of the number of 53BP1 foci per nucleus, and (C) mean volume of 53BP1 foci (*n* > 30 cells per group/mice; 3 mice per genotype; mean ± SEM). (D, E) Same as in (B, C), upon overexpression of SIRT7 WT or catalytically inactive point mutant SIRT7-H188Y in WT and *SirT7*<sup>−/−</sup> primary fibroblasts.

F   Detection of endogenous chromatin-bound and nucleoplasmic 53BP1 protein from WT and *SirT7*<sup>−/−</sup> MEFs before and after IR (8 Gy) by Western blot. Histone H3 was used as loading control and tubulin as control for fractionation.

G   Same as F, using 293T-REX cells treated with doxycycline or vehicle to induce SIRT7-HA expression.

H   ChIP-on-break assay. (Top) Schematic of I-SceI substrate construct introduced into HT1080 cells, which contains a single I-SceI site located within a puromycin resistance cassette. Shown along the top are the coordinates for amplicons probed by Q-PCR. (Bottom) 53BP1 enrichment at the indicated loci in scramble control or *SirT7*-knockdown cells. Q-PCR measurements were normalized to input DNA and non-I-SceI-treated samples (no cut) (mean ± SEM; one of two independent experiments shown). (bottom right) Western blot demonstrating efficient knockdown of *SirT7*, with histone H3 as a loading control.

I, J   NHEJ repair assay using GFP expression-based reporter system in *SirT7*-depleted HT1080 cells (SirT7 knockdown) versus control (scramble control). (I) Representative FACS dot plots. (J) Quantitation of (I); mean ± SEM of three independent clones per condition. Values are normalized to control (scramble control + I-SceI) mean.

K, L   Class-switch recombination in splenic B cells from WT and *SirT7*<sup>−/−</sup> mice stimulated with lipopolysaccharides (LPS) and interleukin-4 (IL-4). (K) The switching from IgM to IgG1 (left) and to IgG3 (right) was measured by FACS. (L) Quantitation of (K) showing mean ± SEM of 3–4 samples per genotype. One representative experiment from two is shown.

Data information: *P < 0.05; **P < 0.01; ***P < 0.001 by ANOVA single factor.
Source data are available online for this figure.

repaired with slow kinetics as compared with euchromatin (Murray *et al*, 2012). However, our data do not support the notion that SIRT7 regulation of 53BP1 recruitment to damaged chromatin affects

heterochromatin repair. We did not observe slower repair kinetics of IR-induced DSBs in SIRT7-depleted cells (Fig 4D), or an accumulation of DNA damage at pericentric heterochromatin (Fig 4E and

F), as we would expect for a heterochromatin repair impairment. However, we cannot discard that SIRT7 participates in the repair of heterochromatin out of the pericentric regions, such as facultative heterochromatin.

We also monitored RAD51 focus formation to investigate whether the other main pathway for DSB repair, HR, was also affected upon *SirT7* deletion. RAD51 recombinase is active during S- and G2-phases of the cell cycle when a sister chromatid is available as a template for recombinational DNA repair (Rothkamm *et al*, 2003). We did not find differences in the number of RAD51 foci per nucleus between *SirT7$^{-/-}$* and WT cells before and at all time points analyzed after IR (Appendix Fig S5A and B). Moreover, spontaneous sister chromatid exchange (SCE), which is mainly mediated by HR (Sonoda *et al*, 1999), was similar between WT and *SirT7$^{-/-}$* cells (Appendix Fig S7C–E). We used a HR reporter locus stably integrated in HT1080 cells (Lio *et al*, 2004). This HR assay is based on the recovery of a puromycin resistance gene after successful repair of an I-SceI-induced DSB. Efficiency of HR is measured by cell colony formation. In agreement with our previous experiments, SIRT7 overexpression did not result in a statistically significant change of HR efficiency (Appendix Fig S7F and G). We conclude that the absence of SIRT7 does not lead to a detectable change of HR levels in our *SirT7$^{-/-}$*-derived primary cells.

Nevertheless, our results indicate that 53BP1 levels at DSB are impaired in the absence of SIRT7. Most importantly, the reduced presence of 53BP1 at DSBs was directly dependent on SIRT7 deacetylase activity because viral-mediated transduction of *SirT7$^{-/-}$* MEFs with an active SIRT7, but not with a catalytically inactive SIRT7 point mutant, rescued both the number and the mean volume of 53BP1 foci per nucleus (Fig 5D and E). Consistent with this, the amount of endogenous chromatin-bound 53BP1 from *SirT7$^{-/-}$* MEFs was decreased compared to WT cells, both before and after IR (Fig 5F), and increased upon SIRT7 overexpression (Fig 5G). The amount of 53BP1 bounded to chromatin was further examined by chromatin immunoprecipitation (ChIP) assays followed by Q-PCR using a HT1080 cell line in which a single DSB is induced by the I-SceI meganuclease (Fnu *et al*, 2011) (ChIP-on-break). The relative enrichment of 53BP1 increases at different distances from the induced DSB in both control (scramble) and SIRT7-depleted (SIRT7-knockdown) cells, but to a lesser extent in SIRT7-knockdown cells (Fig 5H). Taken together, our results indicate that SIRT7-depleted cells have a reduced level of 53BP1 bound to chromatin and an impaired recruitment of this protein to damaged DNA.

We next investigated the effect of SIRT7-mediated 53BP1 recruitment on NHEJ activity using functional assays of cellular DNA repair (Seluanov *et al*, 2004). First, we evaluated NHEJ activity by measuring the frequency of random genomic integration of a linearized plasmid using clonogenic assays. Plasmid integration was reduced in SIRT7-KD cells by twofold, and SIRT7 overexpression resulted in a twofold increase compared with controls (Fig EV3D and Appendix Fig S6A). We further investigated the impact of SIRT7 depletion in NHEJ using a GFP expression-based reporter system. NHEJ activity correlates with the number of GFP$^+$ cells measured by FACS, which is reduced in SIRT7-KD cells compared with control cells (Fig 5I and J, and Appendix Fig S6B). These results were not due to transfection efficiency since SIRT7-KD cells were equally transfected as compared with control cells (Appendix Fig S6C and D). In addition, we evaluated immunoglobulin class switching

recombination, a process that relies on end joining, and specifically 53BP1 activity (Bothmer *et al*, 2011). The switching from IgM to IgG1 and IgG3 was measured by FACS in splenic B cells. We observed a modest but significant reduction in IgG1$^+$ and a twofold reduction in IgG3$^+$ *SirT7$^{-/-}$* cells (Fig 5K and L), independent of the level of B-cell proliferation (Appendix Fig S7A and B). Overall, our results indicate that SIRT7 participates in the maintenance of genome integrity by modulating DNA repair, specifically the activity of NHEJ DSB repair, plausibly by modulating the recruitment of 53BP1 to chromatin.

It is known that 53BP1 recruitment to DSBs is mediated by chromatin-specific modifications, such as the presence of H4K20Me2 and the MDC1-RNF8-RNF168 pathway that mediates ubiquitination of H2AK15Ub (Panier & Boulton, 2014). However, the number of MDC1 foci and the recruitment of MDC1 to chromatin after IR-induced DNA damage were similar between WT and *SirT7$^{-/-}$*-derived cells (Fig EV4A–C). Consistent with this, we did not find differences in the levels of total nuclear H2A ubiquitination (H2A-Ub) or K63 polyubiquitination between primary WT and *SirT7$^{-/-}$* cells (Fig EV4F). In order to enhance the analysis of the RNF168-mediated H2AUb, we transfected 293T cells with an H2A-Flag-tagged plasmid in which K119/K118 has been mutated to arginine (R). This H2A mutant eliminates the strong polycomb-mediated H2A-K119 mono-ubiquitination (Wang *et al*, 2004; Mattiroli *et al*, 2012) (Fig EV4G). Once again, we did not detect differences of H2A-Ub levels in the absence of SIRT7. In addition, we measured H4K20me2 enrichment at a locus adjacent to a single I-SceI meganuclease-induced DSB (Fig EV4D). We could detect increased H4K20me2 levels upon DSB induction in both control and cells with SIRT7 knockdown, as has been reported previously (Pei *et al*, 2011). Most important, we did not observe differences of H4K20me2 levels upon SIRT7 depletion compared to control (Fig EV4E). Overall, our results do not support the notion that these pathways of 53BP1 recruitment to damaged chromatin are regulated by SIRT7.

## SIRT7 is recruited to DSB to promote H3K18 deacetylation

SIRT7 NAD$^+$-dependent deacetylase activity selectively targets H3K18Ac, and its depletion does not affect H3K18Ac cellular levels except at the promoters of a specific set of genes (Barber *et al*, 2012). In contrast, our results indicate that global nuclear levels of H3K18Ac were elevated during G1- and S-phase in *SirT7$^{-/-}$* primary fibroblasts compared with WT, as measured by the quantification of H3K18Ac in individual nuclei (Fig 6A and B). Moreover, we observed a rapid increase in H3K18Ac within 15 min after IR followed by a decline until it reached a level similar to steady-state conditions (Fig 6C). H3K18Ac levels were significantly more elevated in KO cells 15 min after IR, suggesting that SIRT7 might be important in regulating the extent of H3K18Ac at DNA damage sites in early steps of DNA repair. To test this, we first examined whether SIRT7 was recruited to loci of DNA damage. We observed that SIRT7 colocalized with γH2AX at sites of laser-induced DNA damage (Fig 6D and Appendix Fig S8B). To strengthen our observation, we also examined the amount of SIRT7 bound to chromatin by using the ChIP-on-break approach, confirming SIRT7 recruitment at damaged sites (Fig 6E).

We next measured the recruitment kinetics of SIRT7 at sites of laser-induced DNA damage using SIRT7-GFP-expressing cells

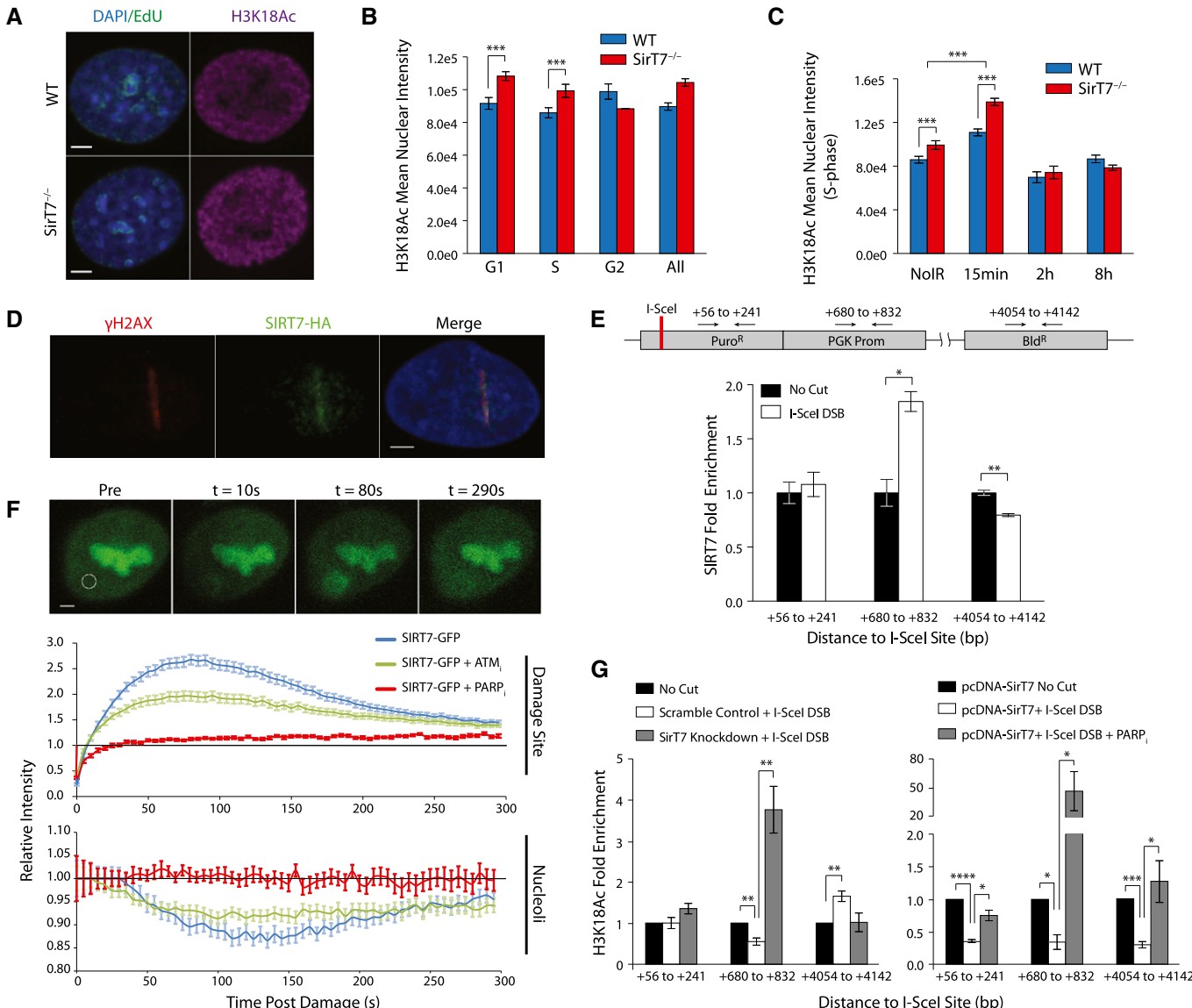

**Figure 6. SIRT7 modulates levels of H3K18Ac at DSB.**

A, B H3K18Ac IF in WT and *SirT7*$^{-/-}$ fibroblasts. Cells were pulsed with EdU (green), stained with antibody against H3K18Ac (magenta), and then counterstained with DAPI (blue). (A) Representative IF images of late S-phase nuclei (EdU positive; scale bar 5 μm). (B) Quantitation of (A) throughout the cell cycle (mean ± SEM of > 30 cells per cell cycle, time point, and genotype; 3 mice per genotype).

C Same as in (B), in S-phase cells at the indicated times after IR (1 Gy) (mean ± SEM of > 30 cells per cell cycle, time point, and genotype; 3 mice per genotype).

D Laser-induced DNA damage in HT1080 cells expressing HA-tagged *SirT7* stained with antibodies against γH2AX (red) and HA (green), then counterstained with DAPI (blue) 30 min postdamage (scale bar 3 μm).

E ChIP assays of SIRT7 enrichment at the indicated loci as described in Fig 5H (mean ± SEM; one of two independent experiments is shown).

F Recruitment kinetics of GFP-tagged *SirT7* after laser-induced microirradiation. (Top) Representative cell at the indicated times after induction of DNA damage (white circle; scale bar 2 μm). (Middle) Quantitation of recruitment kinetics at the site of induced damage in the presence of 5 μM KU-55933 ATM inhibitor (ATM$_i$), 10 μM olaparib PARP inhibitor (PARP$_i$), or DMSO. KU-55933 and olaparib were added 30 min and 1 h, respectively, prior to DNA damage (mean ± SEM; sample size: SIRT7-GFP, *n* = 34; SIRT7-GFP$^+$ ATM$_i$, *n* = 36; SIRT7-GFP$^+$ PARP$_i$ = 18). (Bottom) Same as (middle) except quantitation of SIRT7-GFP relative intensity within the nucleolus. Data were acquired at 5-s intervals over a span of five min.

G ChIP assays of H3K18Ac enrichment at the indicated loci as described in Fig 5H. H3K18Ac enrichment adjacent to the induced DSB in control and SIRT7-depleted cells (bottom left), or in cells overexpressing SIRT7 in the presence or absence of PARP inhibitors (PARP$_i$) (bottom right) (mean ± SEM; one of two independent experiments is shown).

Data information: *$P < 0.05$; **$P < 0.01$; ***$P < 0.001$; ****$P < 0.0001$ by ANOVA single factor.

(Fig 6F, top and middle). SIRT7 begins to accumulate at DNA damage sites ~10 s after the induction of damage ($T_0$ = 9.84 ± 0.05 SEM) and peaks at ~80 s ($\tau_1$ = 36.92 ± 2.81 [95% CI]), followed by

decay in relative intensity ($\tau_2$ = 143.70 ± 5.12 [95% CI]). We observe that SIRT7 remains enriched at the damage site up to 30 min after the induction of damage (Appendix Fig S8E). We

examined the molecular mechanism of SIRT7 recruitment to DSBs. SIRT1 recruitment to damage sites is dependent on ATM activity (Oberdoerffer *et al*, 2008) and correlates with H4K16 hypoacetylation at DNA damage sites (O'Hagan *et al*, 2008). However, our results suggest that ATM activity is not required for the recruitment of SIRT7 to DSBs. The addition of ATM inhibitor (ATM$_i$) did not affect the amount of time required for SIRT7 to begin accumulation at the damage site ($T_0 = 9.84 \pm 0.04$ SEM), though the rate and peak accumulation was marginally reduced ($\tau_1 = 57.23 \pm 1.32$ [95% CI]; $\tau_2 = 129.80 \pm 1.47$ [95% CI]) and is likely attributable to a general depression of DDR (Fig 6F, middle).

We examined whether poly (ADP-ribose) polymerases (PARPs), one of the major NAD$^+$ catabolic enzymes in the cell (Bai & Canto, 2012), affected SIRT7 recruitment to DSB. PARP1, the most ubiquitous and abundant PARP protein, binds chromatin as part of the DNA damage response and is responsible for the recruitment of several DNA damage repair proteins to damaged DNA sites (Krishnakumar & Kraus, 2010). The addition of PARP inhibitor (PARP$_i$) affected all aspects of SIRT7 accumulation at the laser-induced damage site, largely preventing robust accumulation (Fig 6F, middle). Interestingly, we observed a parallel partial depletion of SIRT7-GFP from the nucleolus after the induction of damage (Fig 6F, bottom): The GFP intensity reached a minimum ~135 s after damage induction and was followed by an accumulation phase. Treatment of cells with ATM$_i$ had no effect on this phenomenon, while treatment with PARP$_i$ completely abolished this effect. Control experiments such as FRAP and microirradiation dose–response analysis are in Appendix Fig S8A, C and D. Furthermore, H3K18Ac modulation at damage sites was SIRT7 dependent. The local H3K18Ac level at different distances from a single I-SceI meganuclease-induced DSB indicated that SIRT7 depletion increased H3K18Ac levels by threefold proximal to the DNA break site (Fig 6G, left). Accordingly, overexpression of SIRT7 decreased H3K18Ac levels at the DNA break site, and this effect was reversed in the presence of PARP inhibitors (Fig 6G, right).

**H3K18 deacetylation is necessary for 53BP1 recruitment to DSB**

To further evaluate the impact of H3K18Ac on DSB repair, we performed NHEJ functional assays, similar to the ones described for Figs 5I and J, and EV3D. In this case, we used cells that overexpress histone H3 in which K18 was mutated either to glutamine (K18Q) or to arginine (K18R) to mimic an acetylated or deacetylated lysine residue, respectively (Fig EV5A). NHEJ activity was reduced in the presence of both H3-K18Q and H3-K18R compared with control cells in both the clonogenic random integration assay (Fig 7A) and the GFP-based system (Figs 7B and EV5B). These data indicate that regulation of H3K18Ac levels at DSB sites is necessary for proper NHEJ repair. Furthermore, 53BP1 recruitment was affected by the level of H3K18Ac at DNA damage sites. Both the number and mean volume of 53BP1 foci at IR-induced DSB were reduced in cells carrying the H3-K18Q mutant variant as compared with control (Figs 7C and D, and EV5C). In contrast, the presence of H3-K18R increased the recruitment and volume of foci at DSBs despite the observation that the amount of damaged DNA, as measured by the number of γH2AX foci, was similar between all cell lines (Fig EV5D). These results are consistent with the amounts of chromatin-bound 53BP1 found in these cell lines by Western blot (Fig 7E) and indicate that

53BP1 binding is inhibited by histone H3K18Ac. The amount of the 53BP1 downstream effector, RIF1 (Panier & Boulton, 2014), in the H3K18-mutant cell lines followed the same pattern of reduced recruitment to IR-induced DSB as that observed for 53BP1 (Fig 7F–H) as measured by quantification of foci formation by immunofluorescence. Consistent with this, we also observed reduced RIF1 focus formation at damaged chromatin in *SirT7*$^{-/-}$-derived cells (Fig EV5E). Strikingly, expression of H3K18R mutant in SIRT7-knockdown cells (Fig 7I) and in primary *SirT7*$^{-/-}$ MEFs (Fig 7J) fully rescued chromatin-bound 53BP1 levels measured by ChIP-on-break and Western blot analysis, respectively. Overall, these results further support our identification of H3K18 deacetylation as a novel regulatory pathway for 53BP1 recruitment to chromatin (Fig 7K).

## Discussion

We have demonstrated that genome homeostasis is disrupted in the absence of SIRT7. Our results indicate a novel function of SIRT7 in both embryonic and postnatal development (Fig 1A and Appendix Fig S1D). Sirtuin deficiency has been previously associated with reduced lifespan; for example, *SirT6*$^{-/-}$ mice die within the first month of life with clear signs of accelerated aging (Mostoslavsky *et al*, 2006). However, only SIRT1 deficiency has been associated with severe developmental defects (Cheng *et al*, 2003; McBurney *et al*, 2003; Wang *et al*, 2008a). Therefore, our data expand the role of sirtuins in embryonic development. In addition, we observed an overall decline in the physiological condition of *SirT7*$^{-/-}$ mice, including reduced circulating IGF-1 protein, increased p16INK4 expression, altered HSC compartment, and leukopenia (Fig 2D–H). SIRT1 overexpression in mice results in a delayed onset of age-related pathologies that correlate with reduced p16INK4 expression (Herranz *et al*, 2010). Progeroid mouse models, including *SirT6*$^{-/-}$ mice (Mostoslavsky *et al*, 2006), display a systemic reduction in IGF-1 levels, which might be related to the optimization of energy resources in response to cellular stress. Therefore, the reduced IGF-1 levels in *SirT7*$^{-/-}$ mice may represent an organismal adaptation to chronic DNA damage. In agreement with previous studies (Shin *et al*, 2013; Ryu *et al*, 2014; Cioffi *et al*, 2015), we also observed that SIRT7 deficiency is associated with both reduced visceral fat (Fig 2B and C) and increased hepatic lipid content (Fig EV1A). Defective transcriptional regulation of ribosomal proteins (Shin *et al*, 2013) and of nuclear-encoded mitochondrial genes (Ryu *et al*, 2014) have been proposed to be the underlying mechanism leading to the increased lipid content of *SirT7*$^{-/-}$ hepatocytes. Reduced adipocyte differentiation has also been reported in the white adipose tissue from *SirT7*$^{-/-}$ mice, but the molecular mechanism remains unexplored (Cioffi *et al*, 2015). Loss of white adipose tissue occurs in many progeroid syndromes, but whether this is due to persistent DNA damage has just started to be examined (Karakasilioti *et al*, 2013). Overall, our results provide the first evidence of an accelerated aging phenotype in *SirT7*$^{-/-}$ mice, which is consistent with a previous SIRT7 KO mouse model showing hypertrophic cardiomyopathy and a shortened lifespan (Vakhrusheva *et al*, 2008b).

Here, we provide evidence that the genome instability observed in *SirT7*$^{-/-}$ mice is associated with a functional role of SIRT7 in

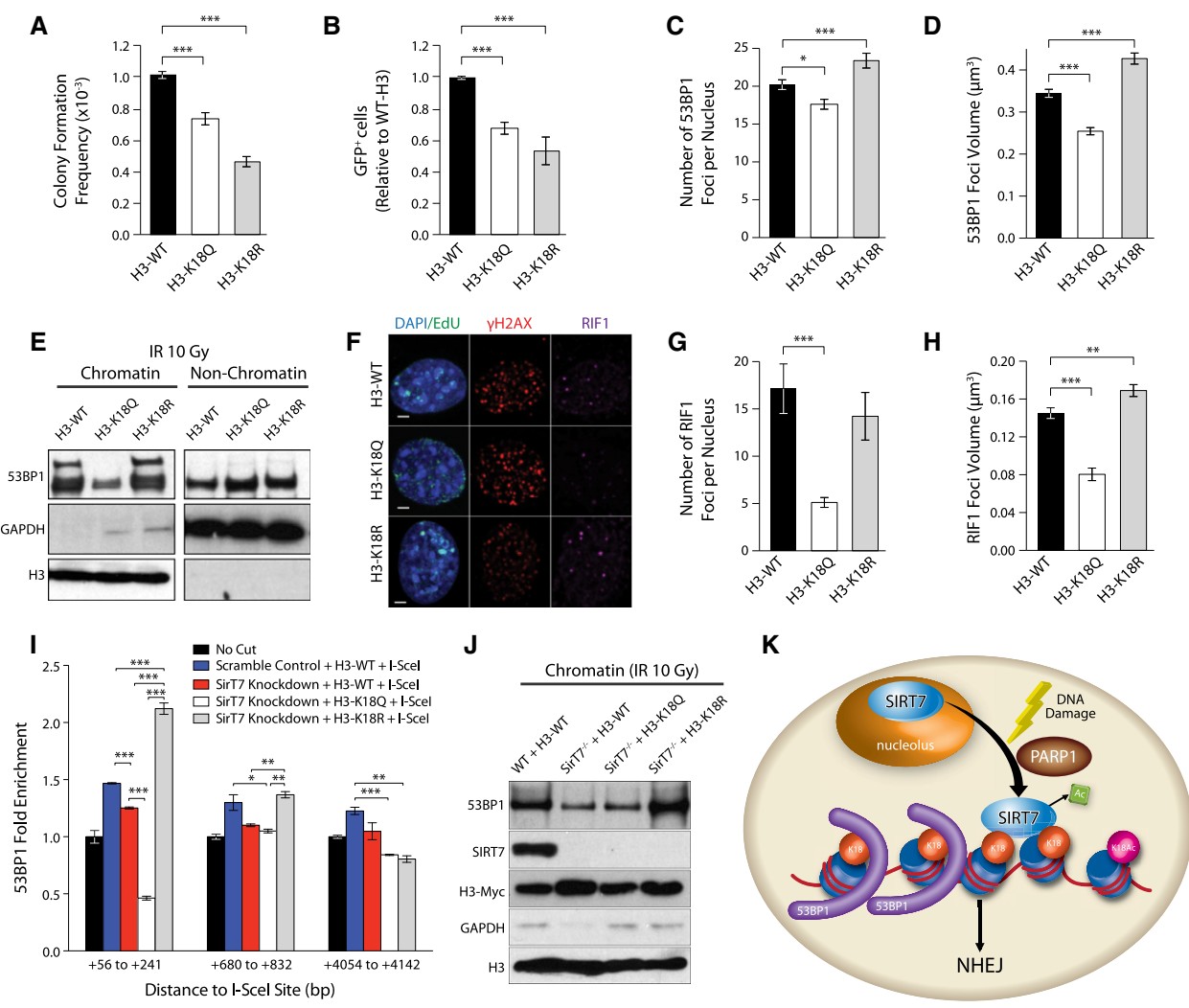

**Figure 7. H3K18 deacetylation is required for efficient 53BP1 recruitment to chromatin.**

NHEJ activity in HT1080 or NIH3T3 cells stably overexpressing H3-WT, H3-K18Q, or H3-K18R (naive, acetylated, or deacetylated H3K18 residues, respectively) (A–H).

A    Random integration assay in HT1080 cells using a linearized pSMCV vector containing a puromycin resistance cassette (mean number of puromycin-resistant colonies normalized by plating efficiency ± SEM; one representative transfection from 3 is shown).

B    NHEJ repair assay using GFP expression-based reporter system described in Fig 5I (mean ± SEM of three independent experiments. See Fig EV5B for representative FACS dot plots).

C, D   IF staining of NIH3T3 cells after IR (1 Gy) and 1-h chase. Cells were pulsed with EdU 30 min prior to fixation, stained with antibodies against γH2AX and 53BP1, and counterstained with DAPI. Representative images can be found in Fig EV5C. (C) Quantitation of the number of 53BP1 foci per nucleus and (D) mean volume of 53BP1 foci (mean ± SEM of > 30 cells per condition and cell cycle stage).

E    Western blot analysis of 53BP1 in chromatin and non-chromatin fractions from NIH3T3 cells expressing H3-WT, H3-K18Q, or H3-K18R vectors and exposed to IR (10 Gy). One representative blot is shown from 3 independent experiments. Shown is 53BP1, GAPDH for fractionation control, and H3 for loading control.

F–H  Same as in (C, D) except using antibodies against γH2AX (red) and RIF1 (magenta). (F) Representative images from cells in S-phase (EdU positive; scale bar 5 μm). (G) Quantitation of the number of RIF1 foci per nucleus, and (H) mean volume of RIF1 foci (mean ± SEM of > 30 cells per condition and cell cycle stage).

I    53BP1 enrichment at the indicated loci as described in Fig 5H in scramble control or *SirT7*-knockdown cells expressing the indicated H3 constructs (mean ± SEM; one of two independent experiments).

J    Western blot analysis of chromatin fractions from primary WT and *SirT7*$^{-/-}$ MEFs expressing the indicated H3 constructs and exposed to IR (10 Gy). Shown are levels of 53BP1, SIRT7, H3-Myc, GAPDH for fractionation control, and H3 for loading control.

K    Model for the role of SIRT7 in NHEJ DNA repair: SIRT7 is recruited to DNA damage sites, where it regulates the extent of histone H3K18 acetylation, which in turn is required for efficient 53BP1 focus formation and overall NHEJ-mediated DNA repair.

Data information: *$P < 0.05$; **$P < 0.01$; ***$P < 0.001$ by ANOVA single factor.
Source data are available online for this figure.

DNA repair. In addition, our results indicate that SIRT7 deficiency leads to replication stress (Figs 4G–I and EV2A–C). Replication stress might be a consequence of impaired NHEJ, as has been

reported previously (Saintigny *et al*, 2001; Lundin *et al*, 2002). However, we cannot rule out that SIRT7 might have an effect on chromatin structure at the replication fork, which has been

suggested for other histone deacetylases (Bhaskara *et al*, 2013; Wells *et al*, 2013), and/or by directly targeting the DNA replication machinery. Therefore, the increase genome instability in SIRT7-depleted cells could be due to a cumulative effect of replication stress and the failure to repair fork-associated DNA damage.

SIRT7 is recruited to DNA damage sites upon genotoxic stress with a concomitant decrease in nucleolar occupancy (Fig 6D–F). It has been reported previously that stress-dependent re-localization of SIRT7 from nucleoli to other nuclear regions leads to a decrease in PolI activity and reduced ribosomal gene transcription (Chen *et al*, 2013). However, whether SIRT7 acquires novel functions under stress conditions remained unexplored. Similar to SIRT7, SIRT1 relocates from major satellite repeats to DNA damage sites with significant consequences for transcriptional regulation (Oberdoerffer *et al*, 2008). Indeed, SIRT1 is an apical effector involved in the stabilization of ATM at DNA breaks, which is essential for DDR propagation (Dobbin *et al*, 2013). SIRT6 also participates in the early events of DDR, as indicated by its rapid accumulation at DNA damage sites, where it recruits the chromatin remodeler SNF2H and focally deacetylates histones H3K56 (Toiber *et al*, 2013) and H3K9 (McCord *et al*, 2009). In addition, SIRT6 depletion impairs the recruitment of downstream effectors from both NHEJ and HR repair pathways to DSB (Kaidi *et al*, 2010; Toiber *et al*, 2013). Our results indicate that SIRT7 is located downstream of SIRT1 and SIRT6 in the DNA damage signaling cascade. Indeed, SIRT7 accumulation peaks at 1 min after laser-induced DNA damage (Fig 6F) in contrast to the extremely early accumulation of SIRT6 and SIRT1 (Dobbin *et al*, 2013; Toiber *et al*, 2013).

Remarkably, SIRT7 recruitment to sites of DNA damage is dependent on PARP protein activity (Fig 6F and G). PARP1 recruitment at laser microirradiation-induced sites of DNA damage (Mortusewicz *et al*, 2007) shows kinetics similar to SIRT7 recruitment (Fig 6F and Appendix Fig S8). A putative connection between sirtuins and PARPs has been hypothesized since the description of sirtuins as deacetylases, as both protein families are $NAD^+$-consuming enzymes (Zhang, 2003). Indeed, under conditions that increase cellular oxidative stress, PARP1-dependent reduction in $NAD^+$ bioavailability impairs SIRT1 activity. On the other hand, PARP2, a much less abundant $NAD^+$ consuming protein, represses SIRT1 activity by directly regulating its transcriptional levels (Canto & Auwerx, 2011). It will be interesting to investigate whether PARP activity affects SIRT7-mediated transcriptional regulation such as the regulation of ribosomal gene expression. On the other hand, SIRT6-mediated mono-ADP ribosylation of PARP1 enhances its activity as a DNA repair mediator (Mao *et al*, 2011). Our results suggests that PARP proteins, at least at the level of DNA repair, are acting upstream of SIRT7 (Fig 6F and G).

Importantly, we demonstrate that SIRT7 focally regulates the extent of H3K18Ac at damage sites (Fig 6G). Our evidence suggests that H3K18Ac is directly involved in DNA repair, since H3K18Ac levels are fine-tuned in response to induced DNA damage (Fig 6C), and overexpression of H3K18 mutants leads to impaired DNA repair due to reduced NHEJ activity (Fig 7A and B). Previous reports have shown a CBP/p300-dependent histone H3 and H4 acetylation, including H3K18Ac, at sites of DNA damage, which facilitates the recruitment of members from the chromatin remodeling SWI/SNF complex (Ogiwara *et al*, 2011). These authors argued that histone acetylation might be involved in the initial steps of DDR by

mediating chromatin relaxation and, in this way, facilitating the accessibility of repair proteins to damaged DNA. H3K18Ac is also involved in genomewide transcriptional regulation (Wang *et al*, 2008b). Therefore, the observed increase in global acetylation levels (Fig 6C) might also been involved in transcriptional remodeling after IR. A key conclusion from our work is that SIRT7-mediated H3K18 deacetylation regulates the recruitment of 53BP1 at DSBs (Fig 7C–H), plausibly by counterbalancing H318Ac levels at sites of DNA damage. Other upstream components in the DDR, such as ATM auto-phosphorylation and γH2AX focus formation at damaged chromatin, are not impaired by SIRT7 loss (Fig 4B–F). Although the recruitment of 53BP1 to DNA damage sites is only partially impaired in the absence of SIRT7, this is not unusual in other models in which chromatin remodeling at DSB is compromised. Some examples include the Suv4-20h histone methyltransferase double-null mice, in which H4K20me2 is reduced (Schotta *et al*, 2008); the SIRT2 KO mice by its impact on the H4K20me1 methyl transferase PRSET7 (Serrano *et al*, 2013).

53BP1 is a key determinant of DSB repair pathway choice and directs DNA repair toward NHEJ by competition with the cell cycle-regulated HR repair protein BRCA1 (Bunting *et al*, 2010). Accordingly, 53BP1 deletion in BRCA1-deficient cells allows replication protein A (RPA) loading and restoration of HR. A previous report suggested that SIRT7 may regulate HR repair (Mao *et al*, 2011). However, the absence of SIRT7 did not lead to a detectable change of HR levels, as measured by RAD51 focus formation (Appendix Fig S5A–C), by measuring SCE events, or by using a HR reporter locus (Appendix Fig S7C–G). Previous observations showed that a 53BP1 mutant lacking the oligomerization domain was still proficient in repressing CtIP-dependent end resection at dysfunctional telomeres (Lottersberger *et al*, 2013), even though NHEJ was affected. Consistent with this, 53BP1 foci volume is remarkably reduced in *SirT7*$^{-/-}$ cells, which suggests an oligomerization impairment of 53BP1 at DNA damage sites. Therefore, it is possible that the remaining 53BP1 observed in *SirT7*$^{-/-}$ cells is still able to promote DNA end protection and inhibit the initiation of HR repair, but leads to compromised NHEJ. On the other hand, we observed increased cell death after X-ray irradiation in the absence of SIRT7 (Fig 3C and D). This suggests the failure of alternative DNA repair pathways to fully compensate for reduced NHEJ activity and argues against a simple competition model between NHEJ and HR pathways.

The current model for 53BP1 recruitment to DNA damage sites mainly involves the recognition of two chromatin marks: H4K20me2 that constitutively interacts with the 53BP1 tudor domain (Botuyan *et al*, 2006) and H2AK15ub via the 53BP1 UDR motif (Fradet-Turcotte *et al*, 2013). These pathways seem not to be affected in the absence of SIRT7, because we did not observe significant differences between WT and SIRT7-deficient cells on the amount of MDC1 foci formation at DSB, and the levels of H2A ubiquitination and H4K20me2 (Fig EV4). Interestingly, histone acetylation has been shown to limit 53PBP1 interaction with chromatin (Tang *et al*, 2013). Increased H4K16Ac levels have been associated with reduced levels of H4K20me2 and therefore reduced 53BP1 recruitment to damaged DNA (Hsiao & Mizzen, 2013; Tang *et al*, 2013). Our results indicate that H3K18Ac also regulates 53BP1 binding to chromatin (Fig 7C–E). It will be interesting to investigate whether this regulation occurs through a direct interaction between 53BP1 and histone H3 deacetylated at K18. Since the 53BP1 tudor

domain seems not to recognize histone H3 tail (Kim *et al*, 2006), other domains should be involved in the direct interaction of H3K18 and 53BP1. Nevertheless, our results show that SIRT7-mediated H3K18 deacetylation at chromatin flanking DSBs constitutes a novel pathway that facilitates the recruitment of 53BP1, which adds to the previously described chromatin features required for efficient 53BP1 binding to sites of DNA damage.

In summary, we demonstrate that loss of SIRT7 reduces embryonic viability and causes premature aging. Although these phenotypes are certainly multifactorial, our results show that one of the underlying mechanisms relies on SIRT7-mediated maintenance of genome integrity. Indeed, our results show that $SirT7^{-/-}$ mice have higher levels of DNA damage, plausibly due to the cumulative effect of replication stress and defective DNA repair, which is in agreement with data showing that unrepaired DNA lesions are strong drivers of human and mouse aging (Lopez-Otin *et al*, 2013). Importantly, we show that SIRT7 mediates 53BP1 binding to chromatin and subsequent efficient NHEJ repair by its deacetylase activity over H3K18Ac at chromatin surrounding DSBs. These findings illuminate a new mechanism by which 53BP1 is recruited to chromatin, and extend our understanding of the contribution of SIRT7 to genome regulation.

# Materials and Methods

Additional details can be found in the Appendix Supplementary Methods.

### Mice

129sv $SirT7^{-/-}$ mice (Shin *et al*, 2013) were backcrossed with C57BL/6 using a speed congenic strategy (Illumina LB Mouse Linkage Panel/Illumina iScan). $Aprt^{+/-}$ mice and $SirT7^{+/-}$ mice in a 129sv genetic background were used to generate $SirT7^{+/+}Aprt^{+/-}$, $SirT7^{+/-}Aprt^{+/-}$, and $SirT7^{-/-}Aprt^{+/-}$ mice. The analysis of the somatic mutations in the *Aprt* gene is described in detail elsewhere (Shao *et al*, 1999, 2000). Animal studies were conducted in accordance with Rutgers University IACUC policies.

### Cell culture and treatments

Mouse ear fibroblasts (fibroblasts) and mouse embryonic fibroblasts (MEFs) were isolated as previously described (Shao *et al*, 1999). Splenic B cells were purified using CD43 MACS beads according to the manufacturer's instructions (Miltenyi Biotec). X-ray irradiation was delivered at the rate of 1 Gy per minute (Faxitron Cabinet X-ray System; Faxitron X-ray Corp.). For hydroxyurea (HU) treatments, cells were treated with 2 mM HU for the indicated periods of time. Pharmacologic inhibition of ATM was performed using the ATM-specific inhibitor, KU-55933 (Abcam), at the dose of 5 μM 30 min prior to DNA damage induction. For PARP1/2 pharmacological inhibition, olaparib (Selleckchem) was used at a dose of 10 μM 1 h prior to laser-induced DNA damage, or after I-SceI transfection when performing ChIP-on-break experiments for 24 h. SA-βGal activity assay was performed according to manufacturer's instructions (Cell Signaling). For all experiments (unless otherwise indicated), primary cells were used between passages 1 and 3.

### Plasmids and cell lines

shRNA-SirT7-pRS (TI306753, TI306755), shRNA-Scr-pRS (TR30003, TR30012), pCMV6-SirT7-GFP (RG205658), and pCMV6-H3.1 (RC214879) plasmids were obtained from Origene. SCR and SIRT7 siRNA were obtained from Sigma and Dharmacon, respectively. Human HA-SirT7 and Flag-SirT7 was cloned into pcDNA4/TO and pcDNA3.1 vectors (Invitrogen). Site-directed mutagenesis was used to generate H3-K18Q, H3-K18R, and SirT7-H188Y mutants. I-SceI-GR-RFP was a gift from Tom Misteli (Addgene plasmid # 17654). H3K18 mutants, SIRT7-Flag, and SIRT7-H188Y-Flag were subcloned into the pMSCVpuro vector and retroviral particles were generated as described previously (Serrano *et al*, 2013).

### NHEJ and HR functional assays

The random integration of foreign DNA and the I-SceI-GFP reporter was used to measure NHEJ repair as described previously (Yeung *et al*, 2012) with some modifications. HR assay was performed as described in Lio *et al* (2004).

### Sister chromatid exchange

Sister chromatid exchange assays were performed as previously described (Misenko & Bunting, 2014) using primary B cells.

### DNA replication fiber assays

DNA fiber spreads were prepared and analyzed by immunofluorescence as previously described in Schwab and Niedzwiedz (2011). Cells were incubated with 25 μM 5-iododeoxyuridine (IdU, Sigma) for 20 min followed by 20-min incubation with 250 μM 5-chlorodeoxyuridine (CldU, Sigma). For analysis of collapse and second origin firing, cells were first incubated with IdU for 20 min, and replication was then blocked with 2 mM of hydroxyurea (HU) for 2 h, followed by washout and release in IdU for 1 h.

### ChIP-on-break assays

ChIP was performed in HT1904 background cells containing a single I-SceI site within a puromycin acetyltransferase gene as previously described (Fnu *et al*, 2011). For H3K18Ac ChIP, cells were transfected with a pCMV-I-SceI plasmid using a Bio-Rad device (Gene Pulser Xcell). For SIRT7 and 53BP1 ChIP, I-SceI-GR-RFP plasmid was used for transfection and cutting was induced by treatment with 1 μM triamcinolone acetonide (Sigma) for 1 h prior to fixation. ChIP was performed as described elsewhere (Chahar *et al*, 2014). Concentration of ChIP and input DNA was measured using Pico-Green (Life Technologies). Quantitative PCR was performed using 1 ng of ChIP and input DNA. Samples were normalized to input and expressed as a ratio relative to non-I-SceI-treated cells. See Appendix Table S1 for primer sequences.

### Imaging

IF assays were carried out as previously described (Serrano *et al*, 2013). Images were acquired using a LSM510 META confocal microscope (Carl Zeiss). 3D reconstruction and image analysis were

performed using Imaris software (Bitplane, A.G.). Microirradiation experiments and quantification of recruitment kinetics were performed following methods previously described (Hable *et al*, 2012) with some modifications. FRAP experiments were carried out as previously described (Bosch-Presegue *et al*, 2011).

### Flow cytometry

Cell cycle and polyploidy analysis was performed as previously described (Serrano *et al*, 2013). For cell survival analysis, thymocytes were stained with Annexin V-FITC (eBiosciences) and 7AAD (Pharmingen) using standard methods. For class switching, splenic B cells were stimulated for 72 h with LPS (25 μg/ml, Sigma) and IL-4 (5 ng/ml, Sigma), then stained with anti-B220-FITC, anti-IgG1-APC. FACS analysis was performed using FACScalibur or a FACS FC500 (Beckman Coulter) and FlowJo 7/8 (Tree Star).

### COMET assay

To detect DSBs, the neutral comet assay was performed as described previously (Olive & Banath, 2006). The images were acquired using a Leica AF-5000 confocal microscope and analyzed and quantified using the Comet Score program (TriTek).

### Measurement of IGF-1 levels

Whole blood was taken via cardiac puncture. Plasma was isolated using standard procedures and stored at −80°C until further use. IGF-1 levels were assayed using an IGF-1 ELSA kit (Abcam) following the manufacturer's protocol.

### Competitive bone marrow transplants

For transplantation studies, we used CD45.1 mice (C57BL/6.SJL-Ptprca, Jackson Laboratories) as recipients. BM cell suspensions were obtained by flushing multiple femurs and tibias of WT and $SirT7^{-/-}$ cells. Following washing and cell counting, $5 \times 10^6$ cells from CD45.2 WT or $SirT7^{-/-}$ cells were mixed with 5 × 106 cells from CD45.1 competitor cells and transferred into irradiated recipients via intravenous tail injection. Eight weeks later, hematopoietic donor cell reconstitution and lineage distribution were evaluated in peripheral blood by FACS, using the CD45.2 marker.

### Statistical analysis

Unless otherwise indicated, results are shown as the mean ± standard error (SEM), with *P*-values calculated by ANOVA single factor (*P*-value < *0.05, **0.01, ***0.001, ****0.0001).

**Expanded View** for this article is available online.

### Acknowledgements
We thank Dr. Danny Reinberg for providing the anti-SIRT7 antibody and Dr. Sara Buonomo for the anti-RIF1 antibody. This study was supported by a grant from the Human Genetics Institute of New Jersey (HGINJ) (to L.S). B.N.V. was supported by postdoctoral fellowships from the Spanish Ministry of Education, Culture, and Sports (EX-2010-278) and the HGINJ. N.S. was supported by predoctoral FPI Fellowship (BES-2012-052200), and AV funded by the Spanish Ministry of Economy and Competitiveness (MINECO) grant SAF2011-25619 and by La Marató de TV3, Spain.

## Author contributions
BNV and LS conceived the study. BNV, JKT, NGS, NK-G, PM-R, and TN developed and performed the experiments. BNV, JKT, SB, AV, and LS analyzed the data. JAT provided financial support and participated in the discussion. BNV, JKT, and LS wrote the manuscript with contribution from the other authors.

## Conflict of interest
The authors declare that they have no conflict of interest.

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
