## [Review Process File · The EMBO Journal]

Manuscript EMBO-2015-93499

SIRT7 promotes genome integrity by regulating non-homologous end joining DNA repair

Berta Vazquez, Joshua Thackray, Nicolas Simonet, Noriko Kane-Goldsmith, Paloma Martinez-Redondo, Trang Nguyen, Samuel Bunting, Alejandro Vaquero, Jay Tischfield and Lourdes Serrano

Corresponding author: Lourdes Serrano, Rutgers University

Review timeline:

Submission date:	16 November 2015
Editorial Decision:	22 December 2015
Revision received:	21 March 2016
Accepted:	27 April 2016

Editor: Hartmut Vodermaier

Transaction Report:

Preliminary Editorial Decision

15 December 2015

Thank you again for submitting your manuscript EMBOJ-2015-93499, "SIRT7 promotes genome integrity by regulating non-homologous end joining DNA repair". We have now received the reports of four expert referees, which I am enclosing copied below. As you will see, the reviewers express interest in the various new findings in your study, yet they also raise a number of substantive concerns regarding the conclusiveness of the study, especially regarding the mechanistic studies and the underlying causal connections between molecular and organismal effects of SIRT7 loss. Before taking a final decision on this manuscript, I would therefore like to give you an opportunity to consider and respond to the referee reports with a brief point-by-point outline on how the major issues might be addressed/clarified; and to comment on the expected feasibility of such experiments as requested by the reviewers. These tentative response (parts of which we may choose to share and discuss with referees) would be taken into account when making our final decision on this manuscript. I would therefore appreciate if you could send us such a response at your earliest convenience, ideally by end of this week or early next week. Should you have any further questions in this regard, please do not hesitate to let me know.

 Referee #1:

In the present MS, Vazquez and colleagues present their work on a new role of SirT7 in DNA repair, which may explain a (previously published) progeroid phenotype of SirT7 deficient mice. Whereas this phenotype was previously published, the mechanism is new. The authors provide an extremely comprehensive case to show that SirT7 is recruited to DNA breaks, where it regulates H3K18 acetylation, where in turn regulates 53BP1 binding and therefore NHEJ. In short, whereas I acknowledge that the amount of work provided is huge, I am not that sure that the mechanism raised can provide an explanation for why Sirt7 deficient mice "age" faster.

Main concern:

As mentioned, the authors' case is that SirT7 regulates 53BP1 binding. Hence, SirT7 deficiency compromises 53BP1 function leading to the observed genomic instability. Whereas the authors have fully demonstrated that SirT7 deficiency leads to genomic instability (which in turn could explain the segmental progeria), the mechanism proposed does not hold. First, 53BP1 functions are only mildly compromised in these mice. The best-known role of 53BP1 is in regulating Class Switch Recombination in B cells, and this is only mildly reduced in SirT7-deficient lymphocytes. In addition, and most importantly, 53BP1 deficient mice do not age prematurely, which essentially ends this case.

As an alternative interpretation, I would suggest that the authors further explore the possibility that SirT7 deficiency is leading to replication stress. In fact, the authors do provide some data in this regard which points towards this direction. Replication Stress can explain the progeroid phenotype and their findings in HSC, increased p16 levels etc... In addition, it is reasonable and likely that an overall change in H3 acetylation levels will challenge DNA replication (HDAC inhibitors such as TSA have this effect). In my view, the authors could simply perform some DNA fiber analyses to explore whether SirT7 deficiency impairs DNA replication, and if so, they could have a mechanism that can explain all of their findings. Otherwise, the one proposed here, even if interesting, cannot be linked to the progeroid phenotype.

Minor concerns

(1) The use of laser protocols as a readout of DNA damage is challenging, since this protocol generates a wide range of stresses that confound the results. Additional methods to confirm the presence of SirT7 and H3K18 deacetylation at break sites would be desirable.

(2) The increase in p16 levels at a young age is considerable. It raises the possibility that the p16 (and the Ink4 locus) might be directly regulated by SirT7. This would be a very important finding, which could also provide the authors with an alternative model to explain the progeria.

Referee #2:

This manuscript proposes that SIRT7 promotes genomic integrity by regulating NHEJ (based on the title).

Specific Comments

1. The authors examine SIRT7^{-/-} mice. These mice were previously characterized (Shin et al, 2013) and were shown to have fatty liver disease. These previous findings are ignored in this manuscript and the Shin et al paper is only mentioned in the context of Materials and Methods. I am curious to know, if the authors observed fatty liver disease in their SIRT7^{-/-} mice.

2. The authors report that SIRT7^{-/-} mice have reduced life span and progeroid features (Figs 1 and 2). These results are interesting and mimic effects described in mice with knockout of the SIRT6 gene.

3. The authors subsequently propose that the progeroid features are explained by a novel role of SIRT7 in promoting DNA repair and genomic integrity. However, this link is not well substantiated. SIRT7 is present at the nucleoli, where it regulates expression of the rDNA genes. Decreased rDNA expression could have indirect effects on aging, on the response to DNA damage and almost on any physiological process. The authors need to demonstrate that the effects of SIRT7 in the DNA damage response are direct. Specifically, to support the title of the manuscript, the authors need to show that SIRT7 has a direct role in NHEJ.

4. The authors use HT1080 cells in which SIRT7 was depleted using siRNA to show that SIRT7 affects cell viability after irradiation (Fig. 3D) and increased senescence (Fig. 2I). Since the authors have access to SIRT7^{-/-} cells, why did they not use these cells to study these phenotypes?

5. Fig. 4D uses γH2AX foci as a DNA DSB marker to examine whether SIRT7 affects DNA repair in G1, S or G2 cells. Using as reference the number of γH2AX foci in non-irradiated cells, the wt and SIRT7^{-/-} cells show equal levels of repair 8 h after irradiation. Thus, based on this assay there is no repair defect.

6. Fig. 4E performs a similar experiment looking at repair of gH2AX foci in euchromatin and heterochromatin in G2 cells. Again, no difference was observed. These results do not support a role of SIRT7 in DNA repair (including NHEJ), as claimed by the authors.
7. Fig. 4G. Differences in S phase arrest after HU treatment could be due to too many different factors (eg. differences in cell cycle kinetics, most likely) and do not imply that SIRT7-deficient cells are more prone to replication stress, as the authors conclude.
8. Fig. 5A is reported to show decreased number of 53BP1 foci after irradiation in SIRT7^{-/-} cells. By looking at the images, it seems to me that there are more 53BP1 foci in the SIRT7^{-/-} cells. Accordingly, I do not have confidence in the results shown in Figs 5B-5E.
9. Fig. 5G and 5H. Both figures show chromatin-bound 53BP1 before and after irradiation, but in different cell types. It seems that 53BP1 behaves somewhat differently in the two cell types. In one case, IR has no effect on 53BP1 chromatin localization; in the other case, IR enhances chromatin localization. Nevertheless, in both cases higher levels of SIRT7 lead to increased 53BP1 chromatin localization. This is interesting.
10. In a GFP-based NHEJ DNA repair assay, depletion of SIRT7 has a good effect. Perhaps, the experiment could be better controlled to show that SIRT7 depletion does not affect GFP transcription and translation. Cotransfection of a plasmid expressing RFP might provide a good internal control.
11. Fig. 6A-C. Quantitating H3K18c signals across different IF slides, as the authors report, is very difficult. How were signal intensities calibrated across the different slides? Additionally, how was the specificity of the antibody validated?
12. Fig. 6D-E. Many proteins localize to laser induced stripes. The very fast on and off kinetics (everything is over after 200 seconds) seem suspicious, since they do not relate to the kinetics of DNA repair.
13. The main effect in Fig. 6F is increased H3K18ac when SIRT7 is depleted. With normal levels of SIRT7, there is much less change in H3K18ac. Since normal cells have SIRT7, it seems that H3K18ac does not change much after induction of DNA DSBs.
14. Fig 7. H3K18Q is not acetylated; it mimicks an acetylated residue (how well, can be debated).
15. Fig. 7E. The authors show decreased 53BP1 chromatin localization in irradiated cells expressing H3K18Q. It would be nice to show also non-irradiated cells. Further, one would expect that expression of H3K18R would rescue 53BP1 chromatin localization in SIRT7^{-/-} cells. Is this true?
16. Fig. 7F. The number of RIF1 foci seems to be the same in all panels. Just the intensity is reduced in the H3K18Q expressing cells. But the intensity of the gH2AX foci is also reduced in these cells, suggesting that variability in staining, since H3K18ac is no likely to affect gH2AX foci. Accordingly, I am not confident in the data shown in panels G and H.

Overall Comment

The authors can address the points raised above, but should also demonstrate that the observed effects are not indirect, for example, following decreased protein synthesis due to a decrease in PolII-mediated transcription.

Referee #3:

In this study, Vazquez, Serrano and colleagues investigate the function of the sirtuin-family deacetylase enzyme SIRT7. They report that SIRT7 loss in mice leads to shortened lifespan and aging-related phenotypes, and that SIRT7 deficient cells show increased genomic instability and defective DNA repair. They also present a series of functional assays in SIRT7 deficient mouse cells or human cell lines that probe the molecular mechanisms of SIRT7 in DNA repair. The authors propose a model in which SIRT7

is recruited to chromatin surrounding DNA damage sites where it deacetylates its substrate H3K18Ac, which in turn regulates association of 53BP1 to DNA DSBs to influence non-homologous end joining (NHEJ) DSB repair. Overall, the study presents interesting and timely analysis of an important enzyme. The linking of SIRT7 to the DNA damage responses is an important finding. At the same time, the authors have tended to overstate some of their conclusions, particularly in cases where the effects of SIRT7 are quite subtle. In addition, some conclusions are not directly supported by the data presented. There are also a number of technical questions that should be resolved. With appropriate revisions, as suggested below, the paper would be well suited for publication at EMBO.

Major concerns:

1. The authors inappropriately conclude causality in several places, e.g. (1) compromised genome integrity in SIRT7-deficient cells "is a consequence of impaired DDR"; (2) impaired NHEJ repair is "due to the lack of SIRT7-mediated H3K18 deacetylation at DNA damage sites," etc. The experiments certainly show that the phenomena are associated with each other, but fall short of establishing causality. More careful wording of such statements should solve this problem. Also, the title should be revised to "SIRT7 promotes genome integrity and regulates non-homologous end joining DNA repair," or something similar that does not conclude causality.

2. The authors state that "both the recruitment and oligomerization of 53BP1 at DSB is impaired in the absence of SIRT7". This appears to be based on immunofluorescence data of numbers of foci and foci volume, and western blots of chromatin bound proteins. Mechanistic conclusions such as "recruitment" and "oligomerization" can't reasonably be made from such data. Here again, the authors should be more attentive to not over-interpreting their data. Moreover, analysis of DSB association would be much more convincing if shown quantitatively by ChIP. Related to this, in western blots of "chromatin-bound" 53BP1 (Figs 5G,H), are biochemical chromatin fractions shown? If so, the full panel of the fractionation should be presented along with controls for the fractionation process in order for chromatin association to be assessed appropriately.

3. In Figure 7, both H3K18 mutations (H3K18Q and H3K18R) reduce colony formation and NHEJ efficiency, even though K18Q is an acetylation mimic whereas H3K18R mimics deacetylation. By contrast, the two mutants have opposite effects on 53BP1 and RIF1. This suggests that different mechanisms may underlie the functional NHEJ results versus the biochemical findings. How do the authors account for this?

In addition, as above, to make conclusions regarding 53BP1 recruitment to DSBs with the different mutants, immunofluorescence studies are not adequate; ChIP assays really are needed to draw such conclusions. Related to this, in 7E, in the westerns of "chromatin-bound" 53BP1 (again controls for the fractionation are needed), the decreased 53BP1 is much more dramatic than for focus formation shown in 7C, D. Does some of the 53BP1 protein decrease occur at other sites (not DSB foci)? ChIP data is important to resolve the differences in the assays.

4. The analysis of H3K18 peptides (acetylated, nonacetylated, methylated) binding to 53BP1 is very preliminary, and are not needed for the central points of this study. The data in Figure 7I,J, of peptide binding is pretty weak and seems very preliminary. The data would be better removed from the current study and used to develop a more rigorous analyses in a separate paper.

Minor concerns:

1. Weight analysis is of female mice only. Were the male results the same?

2. The Kaplan-Meier curve of SIRT7 KO mice looks biphasic. About 20% of the mice die in the first few weeks (very acute), whereas the rest largely survive for many more (14) months. This suggests that separate mechanisms underlie the acute versus adult onset lethality. The authors should better discuss these aspects of the data.

3. SIRT7 KO MEFs do not undergo premature replicative senescence, but KO splenocytes and ear fibroblasts, as well as human SIRT7 knock-down cell lines show increased senescence markers. What might account for this difference?

4. Although ATM and KAP-1 phosphorylation occur in SIRT7 deficient cells, one cannot conclude that the entire DDR is intact, as the authors conclude. They could restate their conclusions to better reflect the specific data.

5. The decrease in H3K18Ac signal in nucleus in Figure 6A seems much more significant than the quantification in 6B. Why?
6. The data in Figure S9A and B could use better labeling; it is hard to figure out what is shown.
7. SIRT7 appears to affect H3K18Ac at distances ~680-832 from the I-SceI- DSB, but not closer to the break. What mechanism do the authors think this reflects?

Referee #4:

This is a manuscript by Serrano and colleagues studying the effects of SIRT7 deletion in the mouse. They observed that SIRT7 deficient mice exhibit a high degree of embryonic and perinatal lethality, and those mice that get to adulthood show a progeroid syndrome. They further demonstrate that lack of SIRT7 in cells causes senescence and genomic instability, both phenotype the authors relate to its H3K18 deacetylase activity and its ability to modulate DNA repair, specifically non-homologous end joining through recruitment of 53BP1 to sites of damage. Although the progeroid syndrome has been described before for SIRT7 deficiency, embryonic lethality and a role in DNA integrity are novel phenotypes, and as such of great interest to the field. The authors did an extensive molecular and biochemical characterization in their studies, and overall the manuscript support most of their hypotheses. However, there are few concerns that if addressed, they will strengthen the manuscript.

Major comments:

- The results regarding the increase number of LSK-positive cells at 4 month-old mice together with the leukopenia and massive increase in p16INK4 mRNA levels is quite intriguing, but has not been followed in detail. Why is the increase in LSK cells observed? An increase in p16 should make these cells to arrest or apoptose, not increase. Is this a compensatory increase? Without performing additional experiments to analyze the functionality of these cells (in vitro differentiation, bone-marrow transplants, etc.) the analysis seems preliminary.
- The replication stress results (Figure 4G) are intriguing, since such an effect for SIRT7 cannot be explained through its putative roles in NHEJ and 53BP1 recruitment. Unless further explored, this observation remains phenomenological.
- The massive decrease in chromatin binding for 53BP1 is striking, and indicates a clear effect of SIRT7 depletion in recruitment of 53BP1 to chromatin. Yet, such results are not consistent with the normal levels of the two known marks recognized by 53BP1 (H4K20me2 and H2AUb). Such results suggest, as the authors claim, that the whole effect is linked to specific inhibition of 53BP1 binding when H3K18 is acetylated. Yet, their results in Figure 7 show only modest effect (~20-30% reduction in repair and ~10% in 53BP1 foci formation) in cells expressing the H3K18Q mutant histones. Furthermore, their binding assay (7I) also showed a modest decrease in binding of 53BP1 to the H3K18Ac peptide following IR. Such results raise questions on whether, mechanistically, K18Ac is sufficient to explain the massive decrease in 53BP1 chromatin binding in the absence of SIRT7.

Minor concerns

- The graph for the defects in class switch recombination (Fig. 5K-L) will be better supported if the original FACs plots are shown. This is important since it represents the only data supporting an in vivo role for SIRT7 in modulating 53BP1-dependent DNA repair.

Author Response to Preliminary Editorial Decision

21 December 2015

Thank you for your willingness to consider for publication our manuscript EMBOJ-2015-93499, "SIRT7 promotes genome integrity by regulating non-homologous end joining DNA repair". First, we would like to thank the reviewers for the thoughtful review of the manuscript and the helpful suggestions. Please see below a point-by-point response to the reviewers concerns. I have also included a list of the new experiments we would carry out if you agree to receive a full revised version of this manuscript.

 RESPONSE TO REFEREES

Referee #1

(Report for Author)

In the present MS, Vazquez and colleagues present their work on a new role of SirT7 in DNA repair, which may explain a (previously published) progeroid phenotype of SirT7 deficient mice. Whereas this phenotype was previously published, the mechanism is new. The authors provide an extremely comprehensive case to show that SirT7 is recruited to DNA breaks, where it regulates H3K18 acetylation, where in turn regulates 53BP1 binding and therefore NHEJ. In short, whereas I acknowledge that the amount of work provided is huge, I am not that sure that the mechanism raised can provide an explanation for why Sirt7 deficient mice "age" faster.

The present study is of broad interest and represents a conceptual advance at several levels: First, it provides evidence for a role for SIRT7 protein in embryonic development that has not previously been reported. Although reduced life span in the SirT7^{-/-} mice was previously reported (Vakhrusheva et al, 2008), this study did not characterize an accelerated aging phenotype. We believe that the present manuscript provides for the first time extensive molecular and phenotypic evidences of accelerated aging.

We agree with the reviewer that accelerated aging cannot be solely explained by SIRT7 regulation of DNA repair. We acknowledge that the strong SIRT7 KO mice phenotype is a consequence of multiple mechanisms. However, we believe we can conclude that part of the explanation for the SirT7^{-/-} mice phenotype is the increased genome instability and impaired DNA damage response that we documented. We apologized if our claims appear to be overstated. In agreement with the suggestion of Reviewer #3 we are willing to change the title of the manuscript and reflect throughout the manuscript the pleiotropic nature of our phenotype. In this sense, we believe that this work will open entirely new lines of future research in the field.

Main concern:

As mentioned, the authors' case is that SirT7 regulates 53BP1 binding. Hence, SirT7 deficiency compromises 53BP1 function leading to the observed genomic instability. Whereas the authors have fully demonstrated that SirT7 deficiency leads to genomic instability (which in turn could explain the segmental progeria), the mechanism proposed does not hold. First, 53BP1 functions are only mildly compromised in these mice. The best-known role of 53BP1 is in regulating Class Switch Recombination in B cells, and this is only mildly reduced in SirT7-deficient lymphocytes. In addition, and most importantly, 53BP1 deficient mice do not age prematurely, which essentially ends this case.

We agree with the reviewer that the recruitment of 53BP1 to DNA damage site is it only partially impaired in the absence of SIRT7. Note that in other models in which chromatin remodeling at DSB is compromised the impact on 53BP1 function is also only partial. Some examples include: the Suv4-20h histone methyltransferase double-null mice, in which H4K20me2 is reduced (Schotta et al., 2008); the SIRT2 KO mice by its impact on the H4K20me1 methyl transferase PRSET7 (Serrano et al., 2013). Nevertheless, the partial impact on recruitment of 53BP1 to damaged DNA in both the absence of SIRT7 and in the presence of the constitutively acetylated H3K18 mutant has a substantial impact on NHEJ activity (60% reduction) as shown in Figures 5H-5I, S6A and 7A-7B. Nevertheless, we agree that SIRT7 might have other effects beyond 53BP1 regulation and we will expand the discussion of this issue accordingly.

As an alternative interpretation, I would suggest that the authors further explore the possibility that SirT7 deficiency is leading to replication stress. In fact, the authors do provide some data in this regard which points towards this direction. Replication Stress can explain the progeroid phenotype and their findings in HSC, increased p16 levels etc... In addition, it is reasonable and likely that an overall change in H3 acetylation levels will challenge DNA replication (HDAC inhibitors such as TSA have this effect). In my view, the authors could simply perform some DNA fiber analyses to explore whether SirT7 deficiency impairs DNA replication, and if so, they could have a mechanism that can explain all of their findings. Otherwise, the one proposed here, even if interesting, cannot be linked to the progeroid phenotype.

We agree with the reviewer that replication stress would have an important impact on genome stability and it could contribute to explain the observed phenotype. We are willing to do the suggested experiment in primary MEFs from WT and SirT7^{-/-} mice to further prove the direct function of SIRT7 activity in

replicative stress by perform rescue experiments similarly to those described in Figures 5D and E. Depending of the results we could extend similar analysis to the H3K18 mutant cell lines. Beyond that point, we believe that probing deeper into this molecular mechanism would be out of the scope of this already large manuscript.

Minor concerns

(1) The use of laser protocols as a readout of DNA damage is challenging, since this protocol generates a wide range of stresses that confound the results. Additional methods to confirm the presence of SirT7 and H3K18 deacetylation at break sites would be desirable.

We already provided evidence of the impact of SIRT7 depletion on H3K18Ac levels at the DNA damage sites induced it by I-SceI endonuclease using ChIP on the break (Figure 6F). We believe that recruitment of DNA repair proteins by both laser induced DNA damage or X-ray- induced foci accumulation are well established methodologies. However, we are willing to assay both SIRT7 and 53BP1 recruitment at I-SceI induced DNA damage by ChIP experiments.

(2) The increase in p16 levels at a young age is considerable. It raises the possibility that the p16 (and the Ink4 locus) might be directly regulated by SirT7. This would be a very important finding, which could also provide the authors with an alternative model to explain the progeria.

In agreement with the reviewer, previous reports (Barber et al., 2012) and our unpublished ChIP-seq assays indicate that SIRT7 is involved in transcriptional regulation genome-wide (manuscript in preparation). In the aforementioned analysis, SIRT7 does not bind to the p16 promoter or known regulatory regions (analysis performed in human K562 and MEFs). However, we cannot rule out that SIRT7 may regulate p16 protein level in a cell type or cell developmental state manner. Most important, other markers of senescence are also upregulated in SIRT7 depleted cells, and the effect is not unique to p16 protein regulation.

Referee #2

(Report for Author)

This manuscript proposes that SIRT7 promotes genomic integrity by regulating NHEJ (based on the title).
Specific Comments

1. The authors examine SIRT7^{-/-} mice. These mice were previously characterized (Shin et al, 2013) and were shown to have fatty liver disease. These previous findings are ignored in this manuscript and the Shin et al paper is only mentioned in the context of Materials and Methods. I am curious to know, if the authors observed fatty liver disease in their SIRT7^{-/-} mice.

We have not analyzed fatty liver disease in SirT7^{-/-} mice. However, we will perform Red Oil O staining in frozen sections from livers of WT and SirT7^{-/-} animals from our tissue bank. In addition, we will include the paper of Shin et al. in the discussion section in relation to the lipodystrophy observed in SirT7^{-/-} mice.

2. The authors report that SIRT7^{-/-} mice have reduced life span and progeroid features (Figs 1 and 2). These results are interesting and mimic effects described in mice with knockout of the SIRT6 gene.

The Reviewer makes an interesting comparison. SIRT6 phenotype is more severe as SirT6^{-/-} mice die within the first month of life (Mostoslavsky et al, 2006). Both proteins participate in the DNA repair response at different levels as discussed in the manuscript.

3. The authors subsequently propose that the progeroid features are explained by a novel role of SIRT7 in promoting DNA repair and genomic integrity. However, this link is not well substantiated. SIRT7 is present at the nucleoli, where it regulates expression of the rDNA genes. Decreased rDNA expression could have indirect effects on aging, on the response to DNA damage and almost on any physiological process. The authors need to demonstrate that the effects of SIRT7 in the DNA damage response are direct. Specifically, to support the title of the manuscript, the authors need to show that SIRT7 has a direct role in NHEJ.

As acknowledge in the introductory paragraph of Reviewer #1, we believe that we already demonstrated that SIRT7 is recruited to DNA breaks, where it regulates H3K18 acetylation, and in turn regulates 53BP1 binding and therefore NHEJ. Indeed, the absence of SIRT7 and the presence of the constitutively acetylated H3K18 mutant have a substantial impact on NHEJ activity (60% reduction) Figures 5I-5J, S6 and 7A-7B. The proof that this effect is direct and requires SIRT7 catalytic activity comes from our rescue experiments shown in Figures 5I-5J. However, to further satisfy the reviewer we can perform rescue experiments using the NHEJ functional assays as described in Figure 5D-5E. As mentioned to Reviewer #1, we will perform additional ChIP experiments to further demonstrate SIRT7 and 53BP1 recruitment at I-SceI-induced DNA damage by ChIP experiments. Please also note our reply to the second Reviewer's "Overall Comment".

4. The authors use HT1080 cells in which SIRT7 was depleted using siRNA to show that SIRT7 affects cell viability after irradiation (Fig. 3D) and increased senescence (Fig. 2I). Since the authors have access to SIRT7^{-/-} cells, why did they not use these cells to study these phenotypes?

No specific reason we did not perform the colony formation assay in primary MEFs. We have several new generated primary MEFs that can be used for that propose. The SA-βGal activity assay is not suitable for primary SirT7^{-/-} cells as the targeting vector to generate SirT7^{-/-} mice contains the bacterial βGal gene (Supplemental Figure 1A).

5. Fig. 4D uses γH2AX foci as a DNA DSB marker to examine whether SIRT7 affects DNA repair in G1, S or G2 cells. Using as reference the number of γH2AX foci in non-irradiated cells, the wt and SIRT7^{-/-} cells show equal levels of repair 8 h after irradiation. Thus, based on this assay there is no repair defect.

Evidenced by the percentage of cell survival and colony formation after IR (figure 3C and 3D), not all the cells survive the IR treatment. Figure 4D reflects the number of γH2AX foci per nucleus, and indicates that the dynamics of repair in those cells that survive X-ray irradiation (IR) is similar in WT and SirT7^{-/-} derived cells. However, differences in the number of foci per nucleus clearly indicate more DNA damage in the absence of SIRT7. In addition, the percentage of cells that show foci, in non-IR conditions (see right), indicates that the number of cells with foci is elevated in the SIRT7KO population, once more suggesting a repair defect. These data will be added to the manuscript.

WHOLE CELL CYCLE	TOTAL CELL #	# CELLS WITH H2AX FOCI	# CELLS WITH 53BP1 FOCI	# CELLS WITH BOTH
WT	258	178	169	169
KO	356	273	177	165
WT	FROM TOTAL CELL #	69%	66%	66%
KO		77%	50%	46%
WT		FROM CELLS WITH H2AX		95%
KO				60%

6. Fig. 4E performs a similar experiment looking at repair of γH2AX foci in euchromatin and heterochromatin in G2 cells. Again, no difference was observed. These results do not support a role of SIRT7 in DNA repair (including NHEJ), as claimed by the authors.

We respectfully disagree with the reviewer's interpretation of figure 4E. While the reviewer is correct that there are no differences in the repair rate of pericentric heterochromatin-associated DSBs, there is clearly a significant difference in the repair of euchromatic-associated DSBs. The primary purpose of this figure is to demonstrate that the DNA repair defect present in SIRT7 deficient cells occurs primarily in euchromatic regions of chromatin, which is consistent with our proposed mechanism in that H3K18Ac is an active chromatin mark that must be deacetylated by SIRT7 for efficient binding of 53BP1.

7. Fig. 4G. Differences in S phase arrest after HU treatment could be due to too many different factors (eg. differences in cell cycle kinetics, most likely) and do not imply that SIRT7-deficient cells are more prone to replication stress, as the authors conclude.

We agree with the Reviewer and we are willing to extend this result by perform DNA fiber analyses to further explore whether SirT7 deficiency impairs DNA replication as stated in the response to the "Main Concern" of Reviewer #1.

8. Fig. 5A is reported to show decreased number of 53BP1 foci after irradiation in SIRT7^{-/-} cells. By looking at the images, it seems to me that there are more 53BP1 foci in the SIRT7^{-/-} cells. Accordingly, I do not have confidence in the results shown in Figs 5B-5E.

We respectfully disagree with the reviewer's assertion that the images in this figure show more 53BP1 foci (magenta, upper right quadrant and 3D rendering in lower right) in SIRT7 deficient cells than WT cells. Furthermore, the images chosen for the figure are representative of the mean number of foci, which was highly reproducible over three independent experiments with greater than 30 nuclei analyzed per genotype/cell cycle/cell line combination. Our only guess as to the how the reviewer came to this conclusion is through a misinterpretation of the figure, mistaking the γ H2AX staining for 53BP1. SIRT7-deficient cells do present more γ H2AX foci (red, lower left quadrant) than WT cells (see data presented in figure 4C-F). If requested, we are prepared to present additional images for the reviewers' scrutiny.

9. Fig. 5G and 5H. Both figures show chromatin-bound 53BP1 before and after irradiation, but in different cell types. It seems that 53BP1 behaves somewhat differently in the two cell types. In one case, IR has no effect on 53BP1 chromatin localization; in the other case, IR enhances chromatin localization. Nevertheless, in both cases higher levels of SIRT7 lead to increased 53BP1 chromatin localization. This is interesting.

We believe that Figure 5G shows an effect on chromatin-bound 53BP1 after irradiation, which we could demonstrate by quantifying the blots using 3 independent experiments. Figures 5G and H not only show different cell lines, but they also represent very different experimental designs and are therefore difficult to compare. Figure 5G reflects steady-state conditions. In Figure 5H doxycycline treatment might exacerbate the DNA damage response, which will add to the effect of overexpression of SIRT7 resulting in a more pronounced effect.

10. In a GFP-based NHEJ DNA repair assay, depletion of SIRT7 has a good effect. Perhaps, the experiment could be better controlled to show that SIRT7 depletion does not affect GFP transcription and translation.

Cotransfection of a plasmid expressing RFP might provide a good internal control.
Transfection of WT and SIRT7 knockdown cells with an RFP vector results in similar RFP expression between cell types. These data are already included in supplemental Figure 6D.

11. Fig. 6A-C. Quantitating H3K18c signals across different IF slides, as the authors report, is very difficult. How were signal intensities calibrated across the different slides? Additionally, how was the specificity of the antibody validated?

Immunofluorescence staining and quantitative imaging analysis is one of our laboratory's expertise and is carried out in a very controlled manner. First, the immunostaining of WT and KO cells is always performed in parallel using the same preparation of all reagents including antibody dilutions. Second, image acquisition is carefully controlled with slides imaged consecutively without system restart using the same acquisition parameters. In addition, we routinely acquire images of reference fluorescent beads over several hours to establish the stability of our imaging lasers and fluorescence yield. Third, image analysis is performed in an unbiased manner using the same segmentation and processing parameters for all images within an experiment. Fourth, all experiments are carried out blind for the operator. Finally, our results are reproducible between independent replicate experiments.

In regards to the specificity of the anti-H3K18Ac antibody obtained from Abcam (#ab1191). In our own hands this antibody fails to recognize our H3 mutant, H3K18R, while specifically recognizing H3WT and H3K18Q variants (Supplementary Figure 10A; mutant H3 variants present a 4KDa shift relative to endogenous H3 due to the presence of a C-Myc tag). In addition, the manufacturer tests every lot of this antibody in a peptide array including peptides for H3 (unmodified), H3K9Ac, H3K14Ac, H3K18Ac, H3K23Ac, H3K27Ac, H3K36Ac, and H4K12Ac. Specificity has also validated by the manufacturer using blocking peptide/antibody incubation followed by western blot experiments. This antibody has been used in over 47 peer-reviewed publications.

12. Fig. 6D-E. Many proteins localize to laser induced stripes. The very fast on and off kinetics (everything is over after 200 seconds) seem suspicious, since they do not relate to the kinetics of DNA repair.

We respectfully disagree with the reviewer on several fronts. First, the fast kinetics described is not unusual for several proteins involved in the DDR including SIRT1 and SIRT6 as discussed in the manuscript. Second, we carried out the same experiment depicted in Figure 6E but extended the

observation time to 30 minutes after the induction of DNA damage (Supplementary Figure 9E). At this time point we continue to observe a relative increase of SIRT7-GFP signal at the induced lesion. Third, we performed the damage experiment using a nuclear localized GFP as a control, in which the GFP failed to localize to the induced lesion. We also performed FRAP experiments using the SIRT7-GFP and GFP alone constructs further supporting the specificity of this assay in detecting SIRT7 recruitment to DNA damage (supplementary Figure 9C, D). Fourth, the re-localization of SIRT7 to the induced lesion can be inhibited specifically by a PARP inhibitor, while the use of ATM inhibitor has little effect. If SIRT7 re-localization was caused nonspecifically by the lesion generation it is extremely unlikely that PARP inhibition would abolish this phenomenon while an ATM inhibitor would have very little effect.

13. The main effect in Fig. 6F is increased H3K18ac when SIRT7 is depleted. With normal levels of SIRT7, there is much less change in H3K18ac. Since normal cells have SIRT7, it seems that H3K18ac does not change much after induction of DNA DSBs.

As stated in the manuscript, we believe that SIRT7-mediated H3K18 deacetylation has a very precise function in the DDR cascade by regulating the recruitment of 53BP1 at DSBs (Figs 7C-7H), plausibly by counterbalancing H3K18Ac levels at sites of DNA damage, which becomes evident when SIRT7 is depleted (Figure 6F, left panel, gray bar).

14. Fig 7. H3K18Q is not acetylated; it mimicks an acetylated residue (how well, can be debated).

The substitution of K for Q residue has been extensively studied and has been widely accepted as a way to mimic acetylation. Our studies with K to Q substitution are not considered alone, but in combination with the other evidence presented.

15. Fig. 7E. The authors show decreased 53BP1 chromatin localization in irradiated cells expressing H3K18Q. It would be nice to show also non-irradiated cells. Further, one would expect that expression of H3K18R would rescue 53BP1 chromatin localization in SIRT7^{-/-} cells. Is this true?

We can definitely do a western blot showing chromatin-bound 53BP1 in non-irradiated conditions. We will perform the suggested rescue experiments in primary cells from SirT7^{-/-} mice.

16. Fig. 7F. The number of RIF1 foci seems to be the same in all panels. Just the intensity is reduced in the H3K18Q expressing cells. But the intensity of the γH2AX foci is also reduced in these cells, suggesting that variability in staining, since H3K18ac is no likely to affect γH2AX foci. Accordingly, I am not confident in the data shown in panels G and H.

We respectfully disagree with the reviewer's conclusion that the images in this figure show the same number of RIF1 foci (magenta) in all cell lines cells. The images chosen for the figure are representative of the mean number of foci, which was highly reproducible over three independent experiments with greater than 30 nuclei analyzed per genotype/cell cycle/cell line combination. We would also like to direct the reviewer to Supplementary Figure 10D showing that γH2AX foci formation is not significantly different between the three H3 variants. However, it appears that during the submission process the images presented in this subpanel suffered from compression artifacts which may contribute to the reviewer's interpretation, and so we intend to replace the images with higher-quality versions of the same nuclei, and we apologize for the poor quality of the submitted image. If requested, we are prepared to present additional images for the reviewers' scrutiny.

Overall Comment

The authors can address the points raised above, but should also demonstrate that the observed effects are not indirect, for example, following decreased protein synthesis due to a decrease in PolII-mediated transcription.

The role of SIRT7 in promoting rDNA transcription is well characterized at the molecular level (Ford et al., 2006, Chen et al., 2013). Indeed, SIRT7 depletion results in reduced rRNA transcription. However, many lines of evidence indicate that reduced ribosome biogenesis is associated with longevity (Lempiainen et al., 2009; Ansborg et al., 2014), and suggests that the positive role of SIRT7 in rDNA transcription cannot account for the segmental progeria observed in SirT7^{-/-} adult mice. In addition, despite the fact that DNA damage inhibits rRNA production (Larsen et al., 2015), whether inhibition of rDNA transcription itself results in DNA damage remains largely unexplored. However, to address the

Reviewer's concern we propose to inhibit Pol-I dependent transcription in WT and SIRT7 deficient cells and analyze γ H2Ax formation and SA- β gal production.

Referee #3

(Report for Author)

In this study, Vazquez, Serrano and colleagues investigate the function of the sirtuin-family deacetylase enzyme SIRT7. They report that SIRT7 loss in mice leads to shortened lifespan and aging-related phenotypes, and that SIRT7 deficient cells show increased genomic instability and defective DNA repair. They also present a series of functional assays in SIRT7 deficient mouse cells or human cell lines that probe the molecular mechanisms of SIRT7 in DNA repair. The authors propose a model in which SIRT7 is recruited to chromatin surrounding DNA damage sites where it deacetylates its substrate H3K18Ac, which in turn regulates association of 53BP1 to DNA DSBs to influence non-homologous end joining (NHEJ) DSB repair. Overall, the study presents interesting and timely analysis of an important enzyme. The linking of SIRT7 to the DNA damage responses is an important finding. At the same time, the authors have tended to overstate some of their conclusions, particularly in cases where the effects of SIRT7 are quite subtle. In addition, some conclusions are not directly supported by the data presented. There are also a number of technical questions that should be resolved. With appropriate revisions, as suggested below, the paper would be well suited for publication at EMBO.

Major concerns:

1. The authors inappropriately conclude causality in several places, e.g. (1) compromised genome integrity in SIRT7-deficient cells "is a consequence of impaired DDR"; (2) impaired NHEJ repair is "due to the lack of SIRT7-mediated H3K18 deacetylation at DNA damage sites," etc. The experiments certainly show that the phenomena are associated with each other, but fall short of establishing causality. More careful wording of such statements should solve this problem. Also, the title should be revised to "SIRT7 promotes genome integrity and regulates non-homologous end joining DNA repair," or something similar that does not conclude causality.

As stated in our response to the Reviewer #1, we apologize if our claims appear to be overstated. We are willing to change the title of the manuscript and to change accordingly our statements.

2. The authors state that "both the recruitment and oligomerization of 53BP1 at DSB is impaired in the absence of SIRT7". This appears to be based on immunofluorescence data of numbers of foci and foci volume, and western blots of chromatin bound proteins. Mechanistic conclusions such as "recruitment" and "oligomerization" can't reasonably be made from such data. Here again, the authors should be more attentive to not over-interpreting their data. Moreover, analysis of DSB association would be much more convincing if shown quantitatively by ChIP.

As stated to previous Reviewers, we are willing to assay both SIRT7 and 53BP1 recruitment at I-SceI induced DNA damage by ChIP experiments. To further support our findings, we will use both SIRT7 knockdown cell lines and H3K18Ac mutants to assay for 53BP1 recruitment at DNA damaged sites.

Related to this, in western blots of "chromatin-bound" 53BP1 (Figs 5G,H), are biochemical chromatin fractions shown? If so, the full panel of the fractionation should be presented along with controls for the fractionation process in order for chromatin association to be assessed appropriately.

We included chromatin fractionation for Figure 7E. We performed similar chromatin fractionation assay for Figures 5G and 5H and we apologize that their omission. We will include controls of fractionation.

3. In Figure 7, both H3K18 mutations (H3K18Q and H3K18R) reduce colony formation and NHEJ efficiency, even though K18Q is an acetylation mimic whereas H3K18R mimics deacetylation. By contrast, the two mutants have opposite effects on 53BP1 and RIF1. This suggests that different mechanisms may underlie the functional NHEJ results versus the biochemical findings. How do the authors account for this?

The Reviewer made an interesting observation, which had already tried to be addressed in the manuscript. H3K18Ac levels are fine-tuned in response to induced DNA damage (Fig 6C). Increase of H3K18Ac might be necessary for proper DNA repair consistent with decreased NHEJ activity on the

H3K18R mutant cell line. As stated in the discussion, previous reports have shown a CBP/p300-dependent histone H3 and H4 acetylation, including H3K18Ac, at sites of DNA damage, which facilitates the recruitment of members from the chromatin remodeling SWI/SNF complex (Ogiwara et al, 2011). Those authors argued that histone acetylation might be involved in the initial steps of DDR by mediating chromatin relaxation and, in this way, facilitating the accessibility of repair proteins to damaged DNA. Nevertheless, our results indicate that SIRT7-mediated H3K18 deacetylation has a very precise function in the DDR cascade by regulating the recruitment of 53BP1 at DSBs (Figs 7C-7H). Consistently, only the H3K18Q mutants failed to properly recruit 53BP1 and downstream factors such as RIF1.

In addition, as above, to make conclusions regarding 53BP1 recruitment to DSBs with the different mutants, immunofluorescence studies are not adequate; ChIP assays really are needed to draw such conclusions.

We already proposed to perform these experiments. Please see response to Reviewer#1, concern #1; Reviewer#2, concern #3; and Reviewer#3, concern#2.

Related to this, in 7E, in the westerns of "chromatin-bound" 53BP1 (again controls for the fractionation are needed)

These data are already included in the manuscript in Supplemental Figure 10E. However we can rearrange the blots and include it in the main Figure.

...the decreased 53BP1 is much more dramatic than for focus formation shown in 7C, D. Does some of the 53BP1 protein decrease occur at other sites (not DSB foci)? ChIP data is important to resolve the differences in the assays.

As noted above, we will perform ChIP experiments in these cells. The X-ray irradiation (IR) doses are very different between experiments, which makes it difficult to directly compare experiments quantitatively. In the western blot (WB) we used 10Gy as it is a much less sensitive assay. By immunofluorescence 10Gy will preclude foci quantification and we used 1Gy. Nevertheless, the WB is just supportive evidence and even if there is not a total correlation, they both go in the same direction. However, it is reasonable to argue that the global levels of 53BP1 are not the same as the 53BP1 actively present at the DSB foci and we cannot exclude that SIRT7 might be modulating 53BP1 protein stability. We believe this is subject of future work.

4. The analysis of H3K18 peptides (acetylated, nonacetylated, methylated) binding to 53BP1 is very preliminary, and are not needed for the central points of this study. The data in Figure 7I,J, of peptide binding is pretty weak and seems very preliminary. The data would be better removed from the current study and used to develop a more rigorous analyses in a separate paper.

We included this data as further evidence for the effect of H3K18Ac on 53BP1 recruitment. We agree with the Reviewer that these results need to be extended and we are currently performing the assays. We are willing to remove these data from this manuscript if the Editor and the rest of Reviewers agree.

Minor concerns:

1. Weight analysis is of female mice only. Were the male results the same?

Yes, we observe the same weight difference phenotype in male mice, and so far we do not observe gender differences in any of the phenotypes analyzed. We will include these data.

2. The Kaplan-Meier curve of SIRT7 KO mice looks biphasic. About 20% of the mice die in the first few weeks (very acute), whereas the rest largely survive for many more (14) months. This suggests that separate mechanisms underlie the acute versus adult onset lethality. The authors should better discuss these aspects of the data.

*The Reviewer made a very interesting observation. As we discussed in response to the "Main Concern" of Reviewer #1, the phenotype of *Sirt7*^{-/-} mice described in this manuscript it is a complex, multifactorial process. However, the accumulation of DNA damage, plausibly as a consequence of replicative stress (also discussed elsewhere above), together with the described DNA repair defect will result in genome instability and lead to embryonic lethality. Consistently, several mouse models of DNA repair deficiencies also suffer from embryonic lethality (Barnes et al., 1998; Xu et al., 2001). In addition, accumulation of DNA damage results in devastating consequences for cellular fitness and cumulatively leads to organism*

aging. Nevertheless, we agree that we cannot rule out that there might be additional mechanisms at play for the acute versus adult onset lethality. We will add this point to the discussion.

3. SIRT7 KO MEFs do not undergo premature replicative senescence, but KO splenocytes and ear fibroblasts, as well as human SIRT7 knock-down cell lines show increased senescence markers. What might account for this difference?

This is again a good observation. Apoptosis versus senescence is a choice made in response to unrepaired DNA lesions, which depends of the cell type. MEFs are more prone to apoptosis versus fibroblasts and splenocytes as terminally differentiated cells are more prone to senescence. However, we are willing to measure MEFs radiation sensitivity by colony formation.

4. Although ATM and KAP-1 phosphorylation occur in SIRT7 deficient cells, one cannot conclude that the entire DDR is intact, as the authors conclude. They could restate their conclusions to better reflect the specific data.

We agree with the Reviewer and we will change our conclusions accordingly.

5. The decrease in H3K18Ac signal in nucleus in Figure 6A seems much more significant than the quantification in 6B. Why?

We double checked the mean intensity of the KO cell and it is slightly over the mean intensity of the population. We apologize and we will change the image to be more representative of the mean data.

6. The data in Figure S9A and B could use better labeling; it is hard to figure out what is shown.

We apologize and we will change it.

7. SIRT7 appears to affect H3K18Ac at distances ~680-832 from the I-SceI- DSB, but not closer to the break. What mechanism do the authors think this reflects?

This is an interesting question. Difficult to conclude but it can be argued that it might be an issue of accessibility due to the presence of other DNA repair proteins at the DSB bare end. The proximal site probed by qPCR (+56 to +241) represents 1-2 nucleosomes from the I-SceI cut site. It could be argued that this position, so close to the break site, is occupied by other DNA repair factors (e.g. Ku70, Ku80, DNA-PKcs, and MRN are thought to be present at the bare ends) and occludes the accessibility of SIRT7 to that site.

Referee #4

(Report for Author)

This is a manuscript by Serrano and colleagues studying the effects of SIRT7 deletion in the mouse. They observed that SIRT7 deficient mice exhibit a high degree of embryonic and perinatal lethality, and those mice that get to adulthood show a progeroid syndrome. They further demonstrate that lack of SIRT7 in cells causes senescence and genomic instability, both phenotypes the authors relate to its H3K18 deacetylase activity and its ability to modulate DNA repair, specifically non-homologous end joining through recruitment of 53BP1 to sites of damage. Although the progeroid syndrome has been described before for SIRT7 deficiency, embryonic lethality and a role in DNA integrity are novel phenotypes, and as such of great interest to the field. The authors did an extensive molecular and biochemical characterization in their studies, and overall the manuscript support most of their hypotheses. However, there are few concerns that if addressed, they will strengthen the manuscript.

Major comments:

The results regarding the increase number of LSK-positive cells at 4 month-old mice together with the leukopenia and massive increase in p16INK4 mRNA levels is quite intriguing, but has not been followed in detail. Why is the increase in LSK cells observed? An increase in p16 should make these cells to arrest or apoptose, not increase. Is this a compensatory increase? Without performing additional experiments to analyze the functionality of these cells (in vitro differentiation, bone-marrow transplants, etc.) the analysis seems preliminary.

We agree with the Reviewer's interpretation. We believe that we are facing a compensatory effect. We did perform the bone marrow transplants experiments that the Reviewer suggests. Our results indicate that SIRT7-deficient cells had a reduced capacity to repopulate the lymphoid compartment compared with WT bone marrow cells. This is part of an ongoing project but we can include it in this manuscript if requested.

The replication stress results (Figure 4G) are intriguing, since such an effect for SIRT7 cannot be explained through its putative roles in NHEJ and 53BP1 recruitment. Unless further explored, this observation remains phenomenological.

As suggested and discussed to Reviewer #1, we are willing to extend this analysis.

The massive decrease in chromatin binding for 53BP1 is striking, and indicates a clear effect of SIRT7 depletion in recruitment of 53BP1 to chromatin. Yet, such results are not consistent with the normal levels of the two known marks recognized by 53BP1 (H4K20me2 and H2AUb). Such results suggest, as the authors claim, that the whole effect is linked to specific inhibition of 53BP1 binding when H3K18 is acetylated. Yet, their results in Figure 7 show only modest effect (~20-30% reduction in repair and ~10% in 53BP1 foci formation) in cells expressing the H3K18Q mutant histones. Furthermore, their binding assay (7I) also showed a modest decrease in binding of 53BP1 to the H3K18Ac peptide following IR. Such results raise questions on whether, mechanistically, K18Ac is sufficient to explain the massive decrease in 53BP1 chromatin binding in the absence of SIRT7.

Experiments in vivo (chromatin binding) and in vitro (pulldowns) cannot be correlated quantitatively, but are consistent qualitatively. Moreover, the H3K18 mutants are competing with endogenous protein levels, which probably reduce the effect. See response to Reviewer 3, "Major Concern #4" for further discussion of the pulldown experiments. As discussed there, we are willing to remove these experiments from the manuscript. Instead, we will use these mutants to measure 53BP1 recruitment at an I-SceI-induced DNA break by ChIP analysis.

Minor concerns

- The graph for the defects in class switch recombination (Fig. 5K-L) will be better supported if the original FACs plots are shown. This is important since it represents the only data supporting an in vivo role for SIRT7 in modulating 53BP1-dependent DNA repair.

We will include it.

List of additional experiments to be performed to satisfy comments of the Reviewers

To further probe the contribution of replication stress to the phenotype observed in SIRT7 deficient cells we will perform DNA fiber analysis in WT and SIRT7 deficient cells before and after overexpressing either WT or catalytically inactive SIRT7. We will also repeat this assay in primary WT cells expressing the H3K18 mutants to question the role of H3K18Ac in this process. We believe this experiment will address the concerns raised by the Reviewer #1 (main concern), Reviewer #2 (concern #7), and Reviewer #4 (major comment #2). We have the necessary cell lines and vectors needed to perform the proposed experiments, and our laboratory has extensive expertise in immunostaining and imaging procedures to perform these assays.

We will use our existing ChIP-at-a-break experimental system to provide further evidence for SIRT7 recruitment to DSBs and demonstrate the impact of SIRT7 loss on 53BP1 accumulation at DSBs. We will immunoprecipitate SIRT7 in WT cells as well as 53BP1 in WT and SIRT7 deficient cells, and cells expressing the H3K18 mutants. We believe these experiments will satisfy the concerns raised by Reviewer #1 (minor concern #1), Reviewer #2 (concern #3), Reviewer #3 (concern #2 and #3), and Reviewer #4 (major concern #3).

We will perform Red Oil O staining in liver sections from WT and KO mice. We believe that this would address the concern raised by Reviewer #2 (concern #1). We already have in our possession the necessary samples preserved and suitable for cryosectioning.

We will repeat the NHEJ-GFP functional assays but this time using the catalytically inactive SIRT7 and WT SIRT7 vectors expressed in WT and SIRT7-deficient cells. We believe that this experiment would address the concern raised by reviewer #2 (concern #3).

We propose to perform western blots probing chromatin bound 53BP1 in primary WT and SIRT7 deficient primary cells expressing the H3K18 mutants. We feel that this experiment would satisfy the concern raised by reviewer #2 (concern #15).

We propose to inhibit Pol-I dependent transcription in WT and KO primary cells and analyze γ H2AX foci by immunofluorescence to satisfy Reviewer #2, overall comment.

We will perform colony formation assays in primary WT and KO MEFs after treatment with increasing doses of X-ray irradiation (as is Figure 3C, 3D). We believe that this experiment would address the concern raised by Reviewer #3 (concern #3).

Cited Literature

Vakhrusheva O, Braeuer D, Liu Z, Braun T, Bober E. Sirt7-dependent inhibition of cell growth and proliferation might be instrumental to mediate tissue integrity during aging. *J Physiol Pharmacol.* 2008 Dec;59 Suppl 9:201-12.

Schotta G, Sengupta R, Kubicek S, Malin S, Kauer M, Callén E, Celeste A, Pagani M, Opravil S, De La Rosa-Velazquez IA, Espejo A, Bedford MT, Nussenzweig A, Busslinger M, Jenuwein T. A chromatin-wide transition to H4K20 monomethylation impairs genome integrity and programmed DNA rearrangements in the mouse. *Genes Dev.* 2008 Aug 1;22(15):2048-61.

Serrano L, Martínez-Redondo P, Marazuela-Duque A, Vazquez BN, Dooley SJ, Voigt P, Beck DB, Kane-Goldsmith N, Tong Q, Rabanal RM, Fondevila D, Muñoz P, Krüger M, Tischfield JA, Vaquero A. The tumor suppressor SirT2 regulates cell cycle progression and genome stability by modulating the mitotic deposition of H4K20 methylation. *Genes Dev.* 2013 Mar 15;27(6):639-53.

Barber MF, Michishita-Kioi E, Xi Y, Tasselli L, Kioi M, Moqtaderi Z, Tennen RI, Paredes S, Young NL, Chen K, Struhl K, Garcia BA, Gozani O, Li W, Chua KF. SIRT7 links H3K18 deacetylation to maintenance of oncogenic transformation. *Nature.* 2012 Jul 5;487(7405):114-8.

Shin J, He M, Liu Y, Paredes S, Villanova L, Brown K, Qiu X, Nabavi N, Mohrin M, Wojnoonski K, Li P, Cheng HL, Murphy AJ, Valenzuela DM, Luo H, Kapahi P, Krauss R, Mostoslavsky R, Yancopoulos GD, Alt FW, Chua KF, Chen D. SIRT7 represses Myc activity to suppress ER stress and prevent fatty liver disease. *Cell Rep.* 2013 Nov 14;5(3):654-65.

Mostoslavsky R, Chua KF, Lombard DB, Pang WW, Fischer MR, Gellon L, Liu P, Mostoslavsky G, Franco S, Murphy MM, Mills KD, Patel P, Hsu JT, Hong AL, Ford E, Cheng HL, Kennedy C, Nunez N, Bronson R, Frendewey D, Auerbach W, Valenzuela D, Karow M, Hottiger MO, Hursting S, Barrett JC, Guarente L, Mulligan R, Demple B, Yancopoulos GD, Alt FW. Genomic instability and aging-like phenotype in the absence of mammalian SIRT6. *Cell.* 2006 Jan 27;124(2):315-29.

Ford E, Voit R, Liszt G, Magin C, Grummt I, Guarente L. Mammalian Sir2 homolog SIRT7 is an activator of RNA polymerase I transcription. *Mammalian Sir2 homolog SIRT7 is an activator of RNA polymerase I transcription.* *Genes Dev.* 2006 May 1;20(9):1075-80.

Chen S, Seiler J, Santiago-Reichert M, Felbel K, Grummt I, Voit R. Repression of RNA polymerase I upon stress is caused by inhibition of RNA-dependent deacetylation of PAF53 by SIRT7. *Mol Cell.* 2013 Nov 7;52(3):303-13.

Lempiäinen H, Shore D. Growth control and ribosome biogenesis. *Curr Opin Cell Biol.* 2009 Dec;21(6):855-63.

Arnsburg K, Kirstein-Miles. Interrelation between protein synthesis, proteostasis and life span. *J. Curr Genomics.* 2014 Feb;15(1):66-75.

Larsen DH, Stucki M. Nucleolar responses to DNA double-strand breaks. *Nucleic Acids Res.* 2015 Nov 28. pii: gkv1312. [Epub ahead of print]

Ogiwara H, Ui A, Otsuka A, Satoh H, Yokomi I, Nakajima S, Yasui A, Yokota J, Kohno T. Histone acetylation by CBP and p300 at double-strand break sites facilitates SWI/SNF chromatin remodeling and the recruitment of non-homologous end joining factors. *Oncogene*. 2011 May 5;30(18):2135-46.

Barnes DE, Stamp G, Rosewell I, Denzel A, Lindahl T. Targeted disruption of the gene encoding DNA ligase IV leads to lethality in embryonic mice. *Curr Biol*. 1998 Dec 17-31;8(25):1395-8.

Xu X, Qiao W, Linke SP, Cao L, Li WM, Furth PA, Harris CC, Deng CX. Genetic interactions between tumor suppressors Brcal and p53 in apoptosis, cell cycle and tumorigenesis. *Nat Genet*. 2001 Jul;28(3):266-71.

1st Editorial Decision

22 December 2015

Thank you for response letter proposing how you may address the referee comments on your current submission, EMBOJ-2015-93499, during the course of a major revision of the study. I have now looked through them and was pleased to see that several important concerns of the referees may in fact be clarified in a relatively straightforward manner. I also realize that the new experiments you propose, in particular the DNA fiber and ChIP-on-break analyses, would potentially be very helpful to address a number of the more major concerns raised. Even though the outcome of these experiments cannot be predicted at this stage, I would therefore like to give you an opportunity to prepare a revised manuscript along the lines suggested, and to resubmit it for our consideration, together with an updated response letter.

With regard to a few open editorial questions:

- I agree that the peptide binding data in Figures 7I/J would either need to be strengthened, removed or de-emphasized (e.g. by moving them into the appendix).
- In cases where you offered to show additional representative images for the referees, I would suggest to include those with the final point-by-point response letter, but also to maybe more clearly label the main figures so as to minimize possible misinterpretation.
- I would appreciate if the earlier Sirt7 mouse model by Shin et al is already mentioned in the Introduction section, together with a brief qualification on how this new study is distinguished from it.
- Finally, I would encourage you to submit source data files for the gel/blot panels shown in several main and appendix figures.

Thank you again for the opportunity to consider this work. I look forward to your revision.

1st Revision - authors' response

21 March 2016

First, we would like to thank the reviewers for the thoughtful review of the manuscript and the helpful suggestions.

Referee #1

(Report for Author)

In the present MS, Vazquez and colleagues present their work on a new role of SirT7 in DNA repair, which may explain a (previously published) progeroid phenotype of SirT7 deficient mice. Whereas this phenotype was previously published, the mechanism is new. The authors provide an extremely comprehensive case to show that SirT7 is recruited to DNA breaks, where it regulates H3K18 acetylation, where in turn regulates 53BP1 binding and therefore NHEJ. In short, whereas I acknowledge that the amount of work provided is huge, I am not that sure that the mechanism raised can provide an explanation for why Sirt7 deficient mice "age" faster.

We appreciate the Reviewer's comments and agree that accelerated aging cannot be solely explained by SIRT7 regulation of DNA repair. We acknowledge that the strong SIRT7 KO mice phenotype is a consequence of multiple mechanisms, including increased replication stress as noted by the Reviewer, and have now expanded on this aspect as requested (see response to main concern # 1). However, we can conclude that part of the explanation for the *Sirt7*^{-/-} mouse phenotype is the increased genome instability and impaired DNA damage response that we have documented. We apologized if our claims appeared to be overstated. In agreement with the suggestion of Reviewer #3, we have changed the title of the manuscript and made modifications throughout to reflect the multifactorial nature of the *Sirt7*^{-/-} phenotype. However, we believe that this work will open entirely new lines of future research in the field, as represents a conceptual advance at several levels: First, it provides evidence for a role for SIRT7 protein in embryonic development that has not previously been reported. Although reduced life span in the *Sirt7*^{-/-} mice was previously reported (Vakhrusheva et al, 2008), this study did not characterize an accelerated aging phenotype. We believe that the present manuscript provides for the first time extensive molecular and phenotypic evidence of accelerated aging.

Main concern:

As mentioned, the authors' case is that SirT7 regulates 53BP1 binding. Hence, SirT7 deficiency compromises 53BP1 function leading to the observed genomic instability. Whereas the authors have fully demonstrated that SirT7 deficiency leads to genomic instability (which in turn could explain the segmental progeria), the mechanism proposed does not hold. First, 53BP1 functions are only mildly compromised in these mice. The best-known role of 53BP1 is in regulating Class Switch Recombination in B cells, and this is only mildly reduced in SirT7-deficient lymphocytes. In addition, and most importantly, 53BP1 deficient mice do not age prematurely, which essentially ends this case.

We agree with the reviewer that the recruitment of 53BP1 to DNA damage sites is it only partially impaired in the absence of SIRT7. Note that in other models in which chromatin remodeling at DSB is

compromised the impact on 53BP1 function is also only partial. Some examples include: the Suv4-20h histone methyltransferase double-null mice, in which H4K20me2 is reduced (Schotta et al, 2008); the SIRT2 KO mice by its impact on the H4K20me1 methyl transferase PRSET7 (Serrano et al, 2013). Nevertheless, the partial impact on recruitment of 53BP1 to damaged DNA in both the absence of SIRT7 and in the presence of the constitutively acetylated H3K18 mutant has a substantial impact on NHEJ activity as shown in Figures 5H-5I, EV3d and 7A-7B. We agree that SIRT7 might have other effects beyond 53BP1 regulation and we have expanded the discussion of this issue accordingly (Please see the response to next Reviewer concern for more discussion).

As an alternative interpretation, I would suggest that the authors further explore the possibility that SirT7 deficiency is leading to replication stress. In fact, the authors do provide some data in this regard which points towards this direction. Replication Stress can explain the progeroid phenotype and their findings in HSC, increased p16 levels etc... In addition, it is reasonable and likely that an overall change in H3 acetylation levels will challenge DNA replication (HDAC inhibitors such as TSA have this effect). In my view, the authors could simply perform some DNA fiber analyses to explore whether SirT7 deficiency impairs DNA replication, and if so, they could have a mechanism that can explain all of their findings. Otherwise, the one proposed here, even if interesting, cannot be linked to the progeroid phenotype.

We agree with the reviewer that replication stress would have an important impact on genome stability and it could contribute to explain the observed progeroid phenotype. We performed the DNA fiber analyses as helpfully suggested by the Reviewer, which now strongly support that SIRT7 deficiency does lead to replication stress (new Fig 4H-I and new EV2A-C). Specifically, loss of SIRT7 resulted in a significant reduction in DNA replication fork velocity in *Sirt7*^{-/-} primary MEFs as compared with WT, which became exacerbated as cells were kept in culture (new Fig 4H-I). Moreover, we observed a remarkable increase in the presence of stalled replication forks upon replication block with hydroxyurea (new EV2A-C), which agrees with our previous analysis included in Fig 4G.

Replication stress might be a consequence of impaired NHEJ as has been reported previously (Lundin et al, 2002; Saintigny et al, 2001). However, and in agreement with the Reviewer's comments, we cannot rule out that SIRT7 might have an effect on chromatin structure at the replication fork, which has been suggested for other histone deacetylases (Bhaskara et al, 2013; Wells et al, 2013), and/or directly targeting the DNA replication machinery. Indeed, SIRT7 might function at two levels: the restart of stalled replication forks and the repair of collapsed replication forks. While future studies should be performed to test this interesting hypothesis, this is out of the scope of this already large manuscript. Nevertheless we agree with the Reviewer that the increase genome instability in SIRT7-depleted cells could be due to a cumulative effect of replication stress and the failure to repair fork-associated DNA damage/DSB, which in turn contributes to the observed accelerated aging in *Sirt7*^{-/-} mice.

Minor concerns

(1) The use of laser protocols as a readout of DNA damage is challenging, since this protocol generates a wide range of stresses that confound the results. Additional methods to confirm the presence of SirT7 and H3K18 deacetylation at break sites would be desirable.

We examined recruitment of DNA repair proteins by both laser induced DNA damage and X-ray- induced foci accumulation, which are well established methodologies. We had already provided evidence of the impact of SIRT7 depletion on H3K18Ac levels at the DNA damage sites induced by I-SceI endonuclease using CHIP-on-break in our first submission (now Figure 6F). As requested by the Reviewer, we performed similar CHIP-on-break analyses to further probe SIRT7 recruitment to DNA damage (new Fig 6F). Note that the presence of SIRT7 at the I-SceI induced DNA break strikingly correlates with H3K18Ac levels (Fig 6G left).

(2) The increase in p16 levels at a young age is considerable. It raises the possibility that the p16 (and the Ink4 locus) might be directly regulated by SirT7. This would be a very important finding, which could also provide the authors with an alternative model to explain the progeria.

In agreement with the reviewer, previous reports (Barber et al, 2012) and our unpublished CHIP-seq assays indicate that SIRT7 is involved in genome-wide transcriptional regulation (manuscript in preparation). In the aforementioned analysis, SIRT7 does not bind to the p16 promoter or known regulatory regions (analysis performed in human K562 and MEFs). However, we cannot rule out that SIRT7 may regulate p16 protein level in a cell type or cell developmental state manner. Most important, other markers of senescence are also upregulated in SIRT7 depleted cells, and the effect is not unique to p16 protein regulation.

Referee #2

(Report for Author)

This manuscript proposes that SIRT7 promotes genomic integrity by regulating NHEJ (based on the title).

Specific Comments

1. The authors examine SIRT7^{-/-} mice. These mice were previously characterized (Shin et al, 2013) and were shown to have fatty liver disease. These previous findings are ignored in this manuscript and the Shin et al paper is only mentioned in the context of Materials and Methods. I am curious to know, if the authors observed fatty liver disease in their SIRT7^{-/-} mice.

To answer the reviewer's question, we have performed Oil Red O staining in livers of WT and *Sirt7*^{-/-} animals to measure hepatic lipid content (new Fig EV1A). Our results are consistent with those observed in Shin et al. 2013. Note that discussion of this manuscript has been incorporated in the Introduction and Discussion sections.

2. The authors report that SIRT7^{-/-} mice have reduced life span and progeroid features (Figs 1 and 2). These results are interesting and mimic effects described in mice with knockout of the SIRT6 gene.

The Reviewer makes an interesting comparison. The SIRT6 phenotype is more severe as *Sirt6*^{-/-} mice die within the first month of life ((Mostoslavsky et al, 2006) . Both proteins participate in the DNA repair response at different levels as discussed in the manuscript.

3. The authors subsequently propose that the progeroid features are explained by a novel role of SIRT7 in promoting DNA repair and genomic integrity. However, this link is not well substantiated. SIRT7 is present at the nucleoli, where it regulates expression of the rDNA genes. Decreased rDNA expression could have indirect effects on aging, on the response to DNA damage and almost on any physiological process. The authors need to demonstrate that the effects of SIRT7 in the DNA damage response are direct. Specifically, to support the title of the manuscript, the authors need to show that SIRT7 has a direct role in NHEJ.

As acknowledged in the introductory paragraph to Reviewer #1, we believe that we have already demonstrated that SIRT7 is recruited to DNA breaks, where it regulates H3K18 acetylation, and in turn regulates 53BP1 binding and therefore NHEJ. Please note that we included new ChIP-on-break analyses to further probe SIRT7 recruitment to DNA damage (new Fig 6F), impaired 53BP1 recruitment in the absence of SIRT7 (new Fig 5K-L), and SIRT7-mediated H3K18 deacetylation effect in 53BP1 recruitment (new Fig 7I). Moreover, in our NHEJ functional assays the absence of SIRT7 and the presence of the constitutively acetylated H3K18 mutant have a substantial impact on NHEJ activity, as indicated by our GFP-based reporter (Fig 5H-I), and colony formation (now Fig EV3D) assays. The proof that this effect is direct and requires SIRT7 catalytic activity comes from our rescue experiments now shown in Fig 5D-E. Please also note our reply to this Reviewer's "Overall Comment".

4. The authors use HT1080 cells in which SIRT7 was depleted using siRNA to show that SIRT7 affects cell viability after irradiation (Fig. 3D) and increased senescence (Fig. 2I). Since the authors have access to SIRT7^{-/-} cells, why did they not use these cells to study these phenotypes?

We thank the Reviewer for pointing out this issue. The SA-βGal activity assay is not suitable for primary *Sirt7*^{-/-} cells as the targeting vector to generate *Sirt7*^{-/-} mice contains the bacterial βGal gene (now Appendix Fig 1A). We already provided evidence of decreased cell viability of *Sirt7*^{-/-} cells using primary thymocytes (Fig 3C).

5. Fig. 4D uses γH2AX foci as a DNA DSB marker to examine whether SIRT7 affects DNA repair in G1, S or G2 cells. Using as reference the number of γH2AX foci in non-irradiated cells, the wt and SIRT7^{-/-} cells show equal levels of repair 8 h after irradiation. Thus, based on this assay there is no repair defect.

As evidenced by the percentage of cell survival and colony formation after X-ray irradiation (IR) (figure 3C and 3D), not all the cells survive the IR treatment. Figure 4D reflects the number of γH2AX foci per nucleus, and indicates that the dynamics of repair in those cells that survive IR is similar in WT and *Sirt7*^{-/-} derived cells.

However, differences in the number of foci per nucleus clearly indicate more DNA damage in the absence of SIRT7. In addition, the percentage of cells that show foci, in non-IR conditions (see right), indicates that the number of cells with foci is elevated in the SIRT7KO population, once more suggesting a repair defect.

WHOLE CELL CYCLE	TOTAL CELL #	# CELLS WITH H2AX FOCI	# CELLS WITH 53BP1 FOCI	# CELLS WITH BOTH
WT	258	178	169	169
KO	356	273	177	165
WT	FROM TOTAL CELL #	69%	66%	66%
KO		77%	50%	46%
WT		FROM CELLS WITH H2AX		95%
KO				60%

6. Fig. 4E performs a similar experiment looking at repair of γ H2AX foci in euchromatin and heterochromatin in G2 cells. Again, no difference was observed. These results do not support a role of SIRT7 in DNA repair (including NHEJ), as claimed by the authors.

We respectfully disagree with the reviewer's interpretation of Figure 4E. While the reviewer is correct that there are no differences in the repair rate of pericentric heterochromatin-associated DSBs, there is clearly a significant difference in the repair of euchromatic-associated DSBs. The primary purpose of this figure is to demonstrate that the DNA repair defect present in SIRT7 deficient cells occurs primarily in euchromatic regions of chromatin. This is consistent with our proposed mechanism in that H3K18Ac is an active chromatin mark that must be deacetylated by SIRT7 for efficient binding of 53BP1.

7. Fig. 4G. Differences in S phase arrest after HU treatment could be due to too many different factors (eg. differences in cell cycle kinetics, most likely) and do not imply that SIRT7-deficient cells are more prone to replication stress, as the authors conclude.

In order to extend our results in this respect, we performed DNA fiber analyses, as suggested by the Reviewer 1, to directly examine replication stress in SIRT7 knockdown cells (new Fig 4H-I and new Fig EV2A-C). Specifically, loss of SIRT7 resulted in a significant reduction of DNA replication fork velocity in *Sirt7*^{-/-} primary MEFs as compared with WT, which became exacerbated as cells were kept in culture (new Fig 4H-I). Moreover, we observed a remarkable increase in the presence of stalled replication forks and the firing of new origins of replication upon replication block with hydroxyurea (new EV2A-C). Our results support that SIRT7 deficiency leads to replication stress, in agreement with the results in Fig 4G. Please see the response to the "Main Concern" of Reviewer #1 for discussion/more details.

8. Fig. 5A is reported to show decreased number of 53BP1 foci after irradiation in SIRT7^{-/-} cells. By looking at the images, it seems to me that there are more 53BP1 foci in the SIRT7^{-/-} cells. Accordingly, I do not have confidence in the results shown in Figs 5B-5E.

We respectfully disagree with the reviewer's assertion that the images in this figure show more 53BP1 foci (magenta, upper right quadrant and 3D rendering in lower right) in SIRT7 deficient cells than WT cells. Furthermore, the images chosen for the figure are representative of the mean number of foci, which was highly reproducible over three independent experiments with greater than 30 nuclei analyzed per genotype/cell cycle/cell line combination. SIRT7-deficient cells do present more γ H2AX foci (red, lower left quadrant) than WT cells (see data presented in figure 4C-F). We apologize and have more clearly labelled the main figures so as to minimize possible misinterpretation (please see Fig 5A). Please see below additional images for the Reviewers' scrutiny.

9. Fig. 5G and 5H. Both figures show chromatin-bound 53BP1 before and after irradiation, but in different cell types. It seems that 53BP1 behaves somewhat differently in the two cell types. In one case, IR has no effect on 53BP1 chromatin localization; in the other case, IR enhances chromatin localization. Nevertheless, in both cases higher levels of SIRT7 lead to increased 53BP1 chromatin localization. This is interesting.

Figures 5G and H (now Figures 5F and 5G, respectively) not only show different cell lines, but they also represent very different experimental designs and are therefore difficult to compare. Figure 5F reflects steady-state conditions. In Figure 5G doxycycline treatment might exacerbate the DNA damage response, which will add to the effect of overexpression of SIRT7 resulting in a more pronounced effect. As the reviewer acknowledged, in both cases higher levels of SIRT7 lead to increased 53BP1 chromatin localization. Please note that these figures now include controls of protein fractionation (chromatin versus nucleoplasmic).

10. In a GFP-based NHEJ DNA repair assay, depletion of SIRT7 has a good effect. Perhaps, the experiment could be better controlled to show that SIRT7 depletion does not affect GFP transcription and translation. Cotransfection of a plasmid expressing RFP might provide a good internal control.

Transfection of WT and SIRT7 knockdown cells with an RFP vector results in similar RFP expression between cell types. These data were already included in supplemental Figure 6C and D (now Appendix Figure S6C and D).

11. Fig. 6A-C. Quantitating H3K18c signals across different IF slides, as the authors report, is very difficult. How were signal intensities calibrated across the different slides? Additionally, how was the specificity of the antibody validated?

Immunofluorescence staining and quantitative imaging analysis is one of our laboratory's expertise and is carried out in a very controlled manner. First, the immunostaining of WT and KO cells is always performed in parallel using the same preparation of all reagents including antibody dilutions. Second, image acquisition is carefully controlled with slides imaged consecutively without system restart using the same acquisition parameters. In addition, we routinely acquire images of reference fluorescent beads over several hours to establish the stability of our imaging lasers and fluorescence yield. Third, image analysis is performed in an unbiased manner using the same segmentation and processing parameters for all images within an experiment. Fourth, all experiments are carried out blind for the operator. Finally, our results are reproducible between independent replicate experiments.

In regards to the specificity of the anti-H3K18Ac antibody obtained from Abcam (#ab1191). In our own hands this antibody fails to recognize our H3 mutant, H3K18R, while specifically recognizing H3WT and H3K18Q variants (Supplementary Figure 10A [now EV5A]; mutant H3 variants present a 4KDa shift relative to endogenous H3 due to the presence of a C-Myc tag). In addition, the manufacturer tests every lot of this antibody in a peptide array including peptides for H3 (unmodified), H3K9Ac, H3K14Ac, H3K18Ac, H3K23Ac, H3K27Ac, H3K36Ac, and H4K12Ac. Specificity has also validated by the manufacturer using blocking peptide/antibody incubation followed by western blot experiments. This antibody has been used in over 47 peer-reviewed publications.

12. Fig. 6D-E. Many proteins localize to laser induced stripes. The very fast on and off kinetics (everything is over after 200 seconds) seem suspicious, since they do not relate to the kinetics of DNA repair.

We respectfully disagree with the reviewer on several fronts. First, the fast kinetics described is not unusual for several proteins involved in the DDR including SIRT1 and SIRT6 as discussed in the manuscript. Second, we carried out the same experiment depicted in Figure 6E but extended the observation time to 30 minutes after the induction of DNA damage (now Appendix Fig 8E). At this time point we continue to observe a relative increase of SIRT7-GFP signal at the induced lesion. Third, we performed the damage experiment using a nuclear localized GFP as a control, in which the GFP failed to localize to the induced lesion. We also performed FRAP experiments using the SIRT7-GFP and GFP alone constructs, further supporting the specificity of this assay in detecting SIRT7 recruitment to DNA damage (now Appendix Fig 8C, D). Fourth, the re-localization of SIRT7 to the induced lesion can be inhibited specifically by a PARP inhibitor, while the use of ATM inhibitor has little effect. If SIRT7 re-localization was caused nonspecifically by the lesion generation it is extremely unlikely that PARP inhibition would abolish this phenomenon while an ATM inhibitor would have very little effect. Furthermore, as mentioned above, we have now included one more layer of evidence that SIRT7 is recruited at DNA damage sites by CHIP-on-break (new Fig 6F).

13. The main effect in Fig. 6F is increased H3K18ac when SIRT7 is depleted. With normal levels of SIRT7, there is much less change in H3K18ac. Since normal cells have SIRT7, it seems that H3K18ac does not change much after induction of DNA DSBs.

As stated in the manuscript, we believe that SIRT7-mediated H3K18 deacetylation has a very precise function in the DDR cascade by regulating the recruitment of 53BP1 at DSBs (Figs 7C-7H), plausibly by counterbalancing H3K18Ac levels at sites of DNA damage, which becomes evident when SIRT7 is depleted (now Fig 6G, left panel, gray bar).

14. Fig 7. H3K18Q is not acetylated; it mimicks an acetylated residue (how well, can be debated).

The Reviewer is correct in the comment. The substitution of K for Q residue has been extensively studied and has been widely accepted as a way to mimic acetylation. Our studies with K to Q substitution are not considered alone, but in combination with the other evidence presented.

15. Fig. 7E. The authors show decreased 53BP1 chromatin localization in irradiated cells expressing H3K18Q. It would be nice to show also non-irradiated cells. Further, one would expect that expression of H3K18R would rescue 53BP1 chromatin localization in SIRT7^{-/-} cells. Is this true?

We agree with the reviewer that it would be interesting to know the effect H3K18 mutants under non-irradiated conditions. However, by western blot analysis we could not get consistent results, which precluded its inclusion in the manuscript. It's important to consider that these overexpression experiments have technical limitations and depend of many factors we cannot control such as the degree of chromatin incorporation of the H3K18 mutants. We believe that only under challenging conditions, such as IR, the specific impact of these cells could stand out. However, we agree with the Reviewer that we cannot rule out that SIRT7 might have an additional role in 53BP1 recruitment beyond H3K18 deacetylation and although interesting, we believe is out of the scope of this manuscript.

Supporting our claims, upon X-ray irradiation-inflicted DNA damage our results are very consistent. To directly answer the reviewer's concern, we have expressed the H3K18R mutant in *Sirt7*^{-/-} primary MEFs and we have analyzed chromatin-bound 53BP1 by Western Blot. H3K18R mutant is able to restore 53BP1 levels in SIRT7 depleted cells similar to those observe in WT cells (new Fig 7J). In addition, we have expressed the H3K18 mutants in Sirt7-depleted cells and we analyzed 53BP1 recruitment by the ChIP-on-break approach (new Fig7I). Taken together, our results demonstrate that SIRT7-mediated H3K18 deacetylation plays a direct role in 53BP1 recruitment.

16. Fig. 7F. The number of RIF1 foci seems to be the same in all panels. Just the intensity is reduced in the H3K18Q expressing cells. But the intensity of the gH2AX foci is also reduced in these cells, suggesting that variability in staining, since H3K18ac is no likely to affect gH2AX foci. Accordingly, I am not confident in the data shown in panels G and H.

We respectfully disagree with the reviewer's conclusion that the images in this figure show the same number of RIF1 foci (magenta) in all cell lines cells. The images chosen for the figure are representative of the mean number of foci, which was highly reproducible over three independent experiments with greater than 30 nuclei analyzed per genotype/cell cycle/cell line combination. We would also like to direct the reviewer to now Fig EV5D showing that γ H2AX foci formation is not significantly different between the three H3 variants. However, it appears that during the submission process the images presented in this subpanel suffered from compression artifacts which may contribute to the reviewer's interpretation, and so we have replaced the images with higher-quality versions, and we apologize for the poor quality of the submitted image. Please see below additional images for the Reviewers' scrutiny.

Overall Comment

The authors can address the points raised above, but should also demonstrate that the observed effects are not indirect, for example, following decreased protein synthesis due to a decrease in PolII-mediated transcription.

The role of SIRT7 in promoting rDNA transcription is well characterized at the molecular level (Chen et al, 2013; Ford et al, 2006). Indeed, SIRT7 depletion results in reduced rRNA transcription. However, many lines of evidence indicate that reduced ribosome biogenesis is associated with longevity (Arnsburg & Kirstein-Miles, 2014; Lempiainen & Shore, 2009). Noteworthy, in several mice models inactivation of key proteins involved in cell growth and metabolism such as mTOR and IGF-1 also results in longevity (Johnson et al, 2015; Schumacher et al, 2008). Overall these reports suggest that the positive role of SIRT7 in rDNA transcription cannot account for the segmental progeria observed in *Sirt7*^{-/-} adult mice. In addition, despite the fact that DNA damage inhibits rRNA production (Larsen & Stucki, 2016), whether inhibition of rDNA transcription itself results in DNA damage remains largely unexplored. Nevertheless, we believed that this updated manuscript version soundly supports the direct role of SIRT7 in DNA repair.

Referee #3

(Report for Author)

In this study, Vazquez, Serrano and colleagues investigate the function of the sirtuin-family deacetylase enzyme SIRT7. They report that SIRT7 loss in mice leads to shortened lifespan and aging-related phenotypes, and that SIRT7 deficient cells show increased genomic instability and defective DNA repair. They also present a series of functional assays in SIRT7 deficient mouse cells or human cell lines that probe the molecular mechanisms of SIRT7 in DNA repair. The authors propose a model in which SIRT7 is recruited to chromatin surrounding DNA damage sites where it deacetylates its substrate H3K18Ac, which in turn regulates association of 53BP1 to DNA DSBs to influence non-homologous end joining (NHEJ) DSB repair. Overall, the study presents interesting and timely analysis of an important enzyme. The linking of SIRT7 to the DNA damage responses is an important finding. At the same time, the authors have tended to overstate some of their conclusions, particularly in cases where the effects of SIRT7 are quite subtle. In addition, some conclusions are not directly supported by the data presented. There are also a number of technical questions that should be resolved. With appropriate revisions, as suggested below, the paper would be well suited for publication at EMBO.

Major concerns:

1. The authors inappropriately conclude causality in several places, e.g. (1) compromised genome integrity in SIRT7-deficient cells "is a consequence of impaired DDR"; (2) impaired NHEJ repair is "due to the lack of SIRT7-mediated H3K18 deacetylation at DNA damage sites," etc. The experiments certainly show that the phenomena are associated with each other, but fall short of establishing causality. More careful wording of such statements should solve this problem. Also, the title should be revised to "SIRT7 promotes genome integrity and regulates non-homologous end joining DNA repair," or something similar that does not conclude causality.

As stated in our response to the Reviewer #1, we apologize if our claims appeared to be overstated. We changed the title of the manuscript to the one suggested by the Reviewer and changed the relevant statements accordingly.

2. The authors state that "both the recruitment and oligomerization of 53BP1 at DSB is impaired in the absence of SIRT7". This appears to be based on immunofluorescence data of numbers of foci and foci volume, and western blots of chromatin bound proteins. Mechanistic conclusions such as "recruitment" and "oligomerization" can't reasonably be made from such data. Here again, the authors should be more attentive to not over-interpreting their data. Moreover, analysis of DSB association would be much more convincing if shown quantitatively by ChIP.

The reviewer is correct that a mechanistic conclusion such as impairment of 53BP1 oligomerization cannot be made from the foci and western blot data alone. However, it is important to consider previously reported observations about 53BP1 foci formation by immunofluorescence, for instance, that 53BP1 oligomerization mutants are able to form IRIF, but the IF signal intensity of the foci that are formed is reduced compared to WT (Lottersberger et al, 2013). This would manifest itself as, and is consistent with, reduced foci volume according to our image analysis methodology.

To strengthen our claims, and as stated in response to previous Reviewers, we performed additional CHIP-on-break studies. These experiments further probed SIRT7 recruitment to DNA damage (new Fig 6F), impaired 53BP1 recruitment in the absence of SIRT7 (new Fig 5K-L), and SIRT7-mediated H3K18 deacetylation effect on 53BP1 recruitment (new Fig 7I).

In our 53BP1 CHIP-on-break experiments we observe that not only is 53BP1 recruitment impaired in *Sirt7* knockdown cells adjacent to the DSB, but 53BP1 levels reach those seen in non-cut conditions at the furthest locus probed, while WT cells continue to display a significant enrichment at that locus. Although we feel that our observations are consistent with an oligomerization defect of 53BP1, we have tone-downed our interpretation of these data in the Discussion section.

Related to this, in western blots of "chromatin-bound" 53BP1 (Figs 5G,H), are biochemical chromatin fractions shown? If so, the full panel of the fractionation should be presented along with controls for the fractionation process in order for chromatin association to be assessed appropriately.

We apologize; protein fractionation controls have been included in now Fig 5F-G.

3. In Figure 7, both H3K18 mutations (H3K18Q and H3K18R) reduce colony formation and NHEJ efficiency, even though K18Q is an acetylation mimic whereas H3K18R mimics deacetylation. By contrast, the two mutants have opposite effects on 53BP1 and RIF1. This suggests that different mechanisms may underlie the functional NHEJ results versus the biochemical findings. How do the authors account for this?

The Reviewer made an interesting observation, which we had already tried to address in the manuscript. H3K18Ac levels are fine-tuned in response to induced DNA damage (Fig 6C). The increase of H3K18Ac might be necessary for proper DNA repair, consistent with decreased NHEJ activity in the H3K18R mutant cell line. As stated in the discussion, previous reports have shown a CBP/p300-dependent histone H3 and H4 acetylation, including H3K18Ac, at sites of DNA damage, which facilitates the recruitment of members from the chromatin remodeling SWI/SNF complex (Ogiwara et al, 2011). Those authors argued that histone acetylation might be involved in the initial steps of DDR by mediating chromatin relaxation and, in this way, facilitating the accessibility of repair proteins to damaged DNA. Nevertheless, our results indicate that SIRT7-mediated H3K18 deacetylation has a very precise function in the DDR cascade by regulating the recruitment of 53BP1 at DSBs (Figs 7C-7H). Consistent with this, only the H3K18Q mutants failed to properly recruit 53BP1 and downstream factors such as RIF1.

In addition, as above, to make conclusions regarding 53BP1 recruitment to DSBs with the different mutants, immunofluorescence studies are not adequate; CHIP assays really are needed to draw such conclusions.

Please see response to this Reviewer concern # 2.

Related to this, in 7E, in the westerns of "chromatin-bound" 53BP1 (again controls for the fractionation are needed)

We apologize again, protein fractionation controls have been included in modified Fig 7E and in new Fig 7J.

...the decreased 53BP1 is much more dramatic than for focus formation shown in 7C, D. Does some of the 53BP1 protein decrease occur at other sites (not DSB foci)? ChIP data is important to resolve the differences in the assays.

The X-ray irradiation (IR) doses are different between experiments, which makes it difficult to directly compare experiments quantitatively. In the western blot (WB) we used 10Gy as it is a much less sensitive assay. By immunofluorescence 10Gy will preclude foci quantification and we used 1Gy. Nevertheless, the WB is just supportive evidence and even if there is not a total correlation, they both change in the same direction. However, it is reasonable to argue that the global levels of 53BP1 are not the same as the 53BP1 actively present at the DSB foci (see response to Reviewer 2 concern # 15 for more discussion).

As we already mentioned to this Reviewer, we have performed the requested ChIP-on-break experiment to further strengthen our conclusions.

4. The analysis of H3K18 peptides (acetylated, nonacetylated, methylated) binding to 53BP1 is very preliminary, and are not needed for the central points of this study. The data in Figure 7I,J, of peptide binding is pretty weak and seems very preliminary. The data would be better removed from the current study and used to develop a more rigorous analyses in a separate paper.

We agree with the Reviewer, and have removed these data from this manuscript.

Minor concerns:

1. Weight analysis is of female mice only. Were the male results the same?

Yes, we observe the same weight difference phenotype in male mice, and so far we do not observe gender differences in any of the phenotypes analyzed. These data are now included in new Appendix Fig 1E.

2. The Kaplan-Meier curve of SIRT7 KO mice looks biphasic. About 20% of the mice die in the first few weeks (very acute), whereas the rest largely survive for many more (14) months. This suggests that separate mechanisms underlie the acute versus adult onset lethality. The authors should better discuss these aspects of the data.

The Reviewer made a very interesting observation. As we discussed in response to the "Main Concern" of Reviewer #1, the phenotype of *Sirt7*^{-/-} mice described in this manuscript it is a complex, multifactorial process. However, the accumulation of DNA damage, plausibly as a consequence of replicative stress (also discussed above), together with the described DNA repair defect will result in genome instability and lead to embryonic lethality. Consistently, several mouse models of DNA repair deficiencies also suffer from embryonic lethality (Hakem, 2008). In addition, accumulation of DNA damage results in devastating consequences for cellular fitness and cumulatively leads to organism aging. Nevertheless,

we agree that we cannot rule out that there might be additional mechanisms at play for the acute versus adult onset lethality.

3. SIRT7 KO MEFs do not undergo premature replicative senescence, but KO splenocytes and ear fibroblasts, as well as human SIRT7 knock-down cell lines show increased senescence markers. What might account for this difference?

We thank the Reviewer for this excellent observation. Our results show that WT and *Sirt7*^{-/-} MEFs undergo similar cell growth arrest (after 6 passages), whereas *Sirt7*^{-/-} splenocytes and ear fibroblasts, as well as human SIRT7 knock-down cell lines show increased senescence markers. This suggests that in response to low amount of genotoxic stress, *Sirt7*^{-/-} MEFs are more prone to apoptosis while *Sirt7*^{-/-} splenocytes and adult fibroblasts are more prone to senescence. We believe that SIRT7 cell type-specific roles may account for this difference.

4. Although ATM and KAP-1 phosphorylation occur in SIRT7 deficient cells, one cannot conclude that the entire DDR is intact, as the authors conclude. They could restate their conclusions to better reflect the specific data.

We agree with the Reviewer and we have changed our conclusions accordingly.

5. The decrease in H3K18Ac signal in nucleus in Figure 6A seems much more significant than the quantification in 6B. Why?

We double checked the mean intensity of the KO cell and it is slightly over the mean intensity of the population. We apologize and we changed the image to be more representative of the mean data.

6. The data in Figure S9A and B could use better labeling; it is hard to figure out what is shown.

We apologize and we have more clearly labelled the now Appendix Figs 8A-B.

7. SIRT7 appears to affect H3K18Ac at distances ~680-832 from the I-SceI- DSB, but not closer to the break. What mechanism do the authors think this reflects?

This is an interesting question. Difficult to conclude but it can be argued that it might be an issue of accessibility due to the presence of other DNA repair proteins at the DSB bare end. The proximal site probed by qPCR (+56 to +241) represents 1-2 nucleosomes from the I-SceI cut site. It could be argued that this position, so close to the break site, is occupied by other DNA repair factors (e.g. Ku70, Ku80, DNA-PKcs, and MRN are thought to be present at the bare ends) and occludes the accessibility of SIRT7 to that site.

Referee #4

(Report for Author)

This is a manuscript by Serrano and colleagues studying the effects of SIRT7 deletion in the mouse. They observed that SIRT7 deficient mice exhibit a high degree of embryonic and perinatal lethality, and those

mice that get to adulthood show a progeroid syndrome. They further demonstrate that lack of SIRT7 in cells causes senescence and genomic instability, both phenotype the authors relate to its H3K18 deacetylase activity and its ability to modulate DNA repair, specifically non-homologous end joining through recruitment of 53BP1 to sites of damage. Although the progeroid syndrome has been described before for SIRT7 deficiency, embryonic lethality and a role in DNA integrity are novel phenotypes, and as such of great interest to the field. The authors did an extensive molecular and biochemical characterization in their studies, and overall the manuscript support most of their hypotheses. However, there are few concerns that if addressed, they will strengthen the manuscript.

Major comments:

The results regarding the increase number of LSK-positive cells at 4 month-old mice together with the leukopenia and massive increase in p16INK4 mRNA levels is quite intriguing, but has not been followed in detail. Why is the increase in LSK cells observed? An increase in p16 should make these cells to arrest or apoptose, not increase. Is this a compensatory increase? Without performing additional experiments to analyze the functionality of these cells (in vitro differentiation, bone-marrow transplants, etc.) the analysis seems preliminary.

We sincerely appreciate the Reviewer's positive assessment of the manuscript. We agree with the Reviewer's interpretation, that we are facing a compensatory effect. Expansion of the LSK population has been previously documented in aged mice, and it is thought to be due in part to their reduced regenerative potential (Sudo et al, 2000). We did perform the competitive bone marrow transplant experiments that the Reviewer suggests. Our results indicate that SIRT7-deficient cells had a reduced capacity to repopulate the lymphoid compartment compared with WT bone marrow cells, and this is now include in new Fig EV1B-C.

- The replication stress results (Figure 4G) are intriguing, since such an effect for SIRT7 cannot be explained through its putative roles in NHEJ and 53BP1 recruitment. Unless further explored, this observation remains phenomenological.

As already included in the response to previous Reviewers, we performed DNA fiber analyses as suggested by the Reviewer 1, to examine replication stress in SIRT7 knockdown cells (new Fig 4H-I and new Fig EV2A-C). Specifically, loss of SIRT7 resulted in a significant reduction in DNA replication fork velocity in *Sirt7*^{-/-} primary MEFs as compared with WT, which became exacerbated as cells were kept in culture (new Fig 4H-I). Moreover, we observed a remarkable increase in the presence of stalled replication forks and the firing of new origins of replication upon replication block with hydroxyurea (new EV2A-C). Our results show that SIRT7 deficiency leads to replication stress, which agrees with the results in Fig 4G. Please see response to the "Main Concern" of Reviewer #1 for discussion/more details.

The massive decrease in chromatin binding for 53BP1 is striking, and indicates a clear effect of SIRT7 depletion in recruitment of 53BP1 to chromatin. Yet, such results are not consistent with the normal levels of the two known marks recognized by 53BP1 (H4K20me2 and H2AUb). Such results suggest, as the authors claim, that the whole effect is linked to specific inhibition of 53BP1 binding when H3K18 is acetylated. Yet, their results in Figure 7 show only modest effect (~20-30% reduction in repair and ~10%

in 53BP1 foci formation) in cells expressing the H3K18Q mutant histones. Furthermore, their binding assay (7I) also showed a modest decrease in binding of 53BP1 to the H3K18Ac peptide following IR. Such results raise questions on whether, mechanistically, K18Ac is sufficient to explain the massive decrease in 53BP1 chromatin binding in the absence of SIRT7.

Experiments *in vivo* (chromatin binding) and *in vitro* (pulldowns) cannot be correlated quantitatively, but are consistent qualitatively (note that the pulldown experiments have been removed for this new version of the manuscript). Our conclusion that H3K18 deacetylation affects 53BP1 recruitment is now strengthened by our rescue experiments of expressing H3K18R mutant in SirT7-depleted cells by ChIP-on-break assay (new Fig 7I), and in primary SirT7^{-/-} MEFs by western blot (Fig 7J). However, it is possible that SIRT7 has an additional role in 53BP1 recruitment beyond H3K18 deacetylation (please see response to Reviewer 2, concern # 15 for more discussion related to this issue).

Minor concerns

- The graph for the defects in class switch recombination (Fig. 5K-L) will be better supported if the original FACs plots are shown. This is important since it represents the only data supporting an *in vivo* role for SIRT7 in modulating 53BP1-dependent DNA repair.

We apologize, the original FACs plots are now included in now Fig EV3E.

Cited Literature

Arnsburg K, Kirstein-Miles J (2014) Interrelation between protein synthesis, proteostasis and life span. *Curr Genomics* **15**: 66-75

Barber MF, Michishita-Kioi E, Xi Y, Tasselli L, Kioi M, Moqtaderi Z, Tennen RI, Paredes S, Young NL, Chen K, Struhl K, Garcia BA, Gozani O, Li W, Chua KF (2012) SIRT7 links H3K18 deacetylation to maintenance of oncogenic transformation. *Nature* **487**: 114-118

Bhaskara S, Jacques V, Rusche JR, Olson EN, Cairns BR, Chandrasekharan MB (2013) Histone deacetylases 1 and 2 maintain S-phase chromatin and DNA replication fork progression. *Epigenetics Chromatin* **6**: 27

Chen S, Seiler J, Santiago-Reichel M, Felbel K, Grummt I, Voit R (2013) Repression of RNA polymerase I upon stress is caused by inhibition of RNA-dependent deacetylation of PAF53 by SIRT7. *Mol Cell* **52**: 303-313

Ford E, Voit R, Liszt G, Magin C, Grummt I, Guarente L (2006) Mammalian Sir2 homolog SIRT7 is an activator of RNA polymerase I transcription. *Genes Dev* **20**: 1075-1080

Hakem R (2008) DNA-damage repair; the good, the bad, and the ugly. *EMBO J* **27**: 589-605

Johnson SC, Sangesland M, Kaerberlein M, Rabinovitch PS (2015) Modulating mTOR in aging and health. *Interdiscip Top Gerontol* **40**: 107-127

Larsen DH, Stucki M (2016) Nucleolar responses to DNA double-strand breaks. *Nucleic Acids Res* **44**: 538-544

Lempiainen H, Shore D (2009) Growth control and ribosome biogenesis. *Curr Opin Cell Biol* **21**: 855-863

Lotterberger F, Bothmer A, Robbiani DF, Nussenzweig MC, de Lange T (2013) Role of 53BP1 oligomerization in regulating double-strand break repair. *Proc Natl Acad Sci U S A* **110**: 2146-2151

Lundin C, Erixon K, Arnaudeau C, Schultz N, Jenssen D, Meuth M, Helleday T (2002) Different roles for nonhomologous end joining and homologous recombination following replication arrest in mammalian cells. *Mol Cell Biol* **22**: 5869-5878

Mostoslavsky R, Chua KF, Lombard DB, Pang WW, Fischer MR, Gellon L, Liu P, Mostoslavsky G, Franco S, Murphy MM, Mills KD, Patel P, Hsu JT, Hong AL, Ford E, Cheng HL, Kennedy C, Nunez N, Bronson R, Frendewey D, Auerbach W, Valenzuela D, Karow M, Hottiger MO, Hursting S, Barrett JC, Guarente L, Mulligan R, Demple B, Yancopoulos GD, Alt FW (2006) Genomic instability and aging-like phenotype in the absence of mammalian SIRT6. *Cell* **124**: 315-329

Ogiwara H, Ui A, Otsuka A, Satoh H, Yokomi I, Nakajima S, Yasui A, Yokota J, Kohno T (2011) Histone acetylation by CBP and p300 at double-strand break sites facilitates SWI/SNF chromatin remodeling and the recruitment of non-homologous end joining factors. *Oncogene* **30**: 2135-2146

Saintigny Y, Delacote F, Vares G, Petitot F, Lambert S, Averbek D, Lopez BS (2001) Characterization of homologous recombination induced by replication inhibition in mammalian cells. *EMBO J* **20**: 3861-3870

Schotta G, Sengupta R, Kubicek S, Malin S, Kauer M, Callen E, Celeste A, Pagani M, Opravil S, De La Rosa-Velazquez IA, Espejo A, Bedford MT, Nussenzweig A, Busslinger M, Jenuwein T (2008) A chromatin-wide transition to H4K20 monomethylation impairs genome integrity and programmed DNA rearrangements in the mouse. *Genes Dev* **22**: 2048-2061

Schumacher B, Garinis GA, Hoeijmakers JH (2008) Age to survive: DNA damage and aging. *Trends Genet* **24**: 77-85

Serrano L, Martinez-Redondo P, Marazuela-Duque A, Vazquez BN, Dooley SJ, Voigt P, Beck DB, Kane-Goldsmith N, Tong Q, Rabanal RM, Fondevila D, Munoz P, Kruger M, Tischfield JA, Vaquero A (2013) The tumor suppressor Sirt2 regulates cell cycle progression and genome stability by modulating the mitotic deposition of H4K20 methylation. *Genes Dev* **27**: 639-653

Sudo K, Ema H, Morita Y, Nakauchi H (2000) Age-associated characteristics of murine hematopoietic stem cells. *J Exp Med* **192**: 1273-1280

Vakhrusheva O, Smolka C, Gajawada P, Kostin S, Boettger T, Kubin T, Braun T, Bober E (2008) Sirt7 increases stress resistance of cardiomyocytes and prevents apoptosis and inflammatory cardiomyopathy in mice. *Circ Res* **102**: 703-710

Wells CE, Bhaskara S, Stengel KR, Zhao Y, Sirbu B, Chagot B, Cortez D, Khabele D, Chazin WJ, Cooper A, Jacques V, Rusche J, Eischen CM, McGirt LY, Hiebert SW (2013) Inhibition of histone deacetylase 3 causes replication stress in cutaneous T cell lymphoma. *PLoS One* **8**: e68915

H3-WT**H3-K18Q****H3-K18R****EdU/DAPI** **γ H2AX****RIF1**
Thank you again for submitting your revised manuscript for our consideration, and please excuse the delay in its re-evaluation. We have in the meantime received comments from three of the original reviewers (please see below), who overall consider the manuscript significantly improved and therefore raise no further principle objections towards publication. We shall therefore be happy to accept your study for publication in The EMBO Journal.

Since referee 1 still retained reservations regarding the conclusions on SIRT7 effects on 53BP1 and NHEJ, I had to read the paper, figures and response letter once more in detail. My conclusion was that the writing of the paper is at this stage sufficiently circumspect and cautious as to not over-interpret the moderate but significant effects that were observed, and that major additional rewriting would not appear warranted. I would however suggest the following minor modifications to title and abstract, which should also help to emphasize some of the key new findings (such as PARP-dependent SIRT7 recruitment) better. Please let me know whether you are happy with these proposed modifications, and we would introduce them into the manuscript from our side prior to acceptance.

The only other minor point, raised by referee 4, concerns to possible better images for the DNA fibre analyses. Since these are only example images for the associated quantification, I will leave this up to - should you want to provide modified figures containing other examples, please simply send them to our office at this stage and again we would replace them from our side.

REFEREE REPORTS

Referee #1

As before, my opinion is that this is a nice manuscript with a huge amount of quality data that documents the presence of genomic instability which could (perhaps) explain the progeroid phenotype of SirT7 deficient mice. However, my main concern still lies on the impact of NHEJ-53BP1 on all this, which the authors still consider an important point (highlighted in the abstract etc).

- Again, the impact of Sirt7 on 53BP1 function is very modest (see CSR data). And, even if it were to be absolute, this does not lead to ageing. 53BP1 deficient mice do not age prematurely, if anything, they have been shown to rescue other detrimental DNA repair mutants such as BRCA1 hypomorphs.

- I must admit that I am skeptical of the Sirtuins playing a direct role at DNA breaks. Prominent works have previously reported direct claims of Sirt1 at DNA breaks which supposedly regulated the DNA damage response. These claims have not been validated by various other laboratories, and have led to the loss of funds and time from many groups. Sirtuins are very pleiotropic enzymes, and for me to be convinced that Sirt7 plays an actual role at the sites of damage I would need more convincing work.

- The good news is that as predicted the authors are able to see increased levels of replication stress in their mice by fiber assays. This is consistent with many of their findings (micronuclei, polyploidy, hypersensitivity of the bone marrow, etc...), and likely suffices to explain the ageing of SirT7 deficient mice. It is possible that a similar mechanism actually accounts for the genomic instability reported in the absence of other Sirtuins, and because of that, if a clear message is sent, this paper would do much good to the community. Bear in mind that the one clear phenotype of sirtuin deficient yeast strains is the accumulation of rDNA circles, which we know accumulate due to replication fork stalling followed by recombination... By the way, NHEJ deficiency can not explain replication stress. It can aggravate the consequences of RS, since this leads to DNA breaks, but it does NOT generate RS.

All in all, I think this is a good manuscript, and deserving visibility; but re-orienting its angle and diminishing the NHEJ aspect could be of much more value for the scientific community. Otherwise, in my opinion, it will keep adding to the incremental confusion of the direct roles of sirtuins at sites of DNA breaks.

Referee #3

In this revision, Serrano and colleagues have done a nice job addressing the previous concerns with both thoughtful discussions and new experiments. Their edited manuscript also better describes their data without overstating. The study is timely and about a very important enzyme. I recommend publication.

Referee #4

In this revised version of their manuscript, Serrano and colleagues did major efforts to address previous reviewers concerns. In particular, the new DNA fiber assays, the new ChIP assays in the Isce-I system, and the rescue of 53BP1 binding upon expression of the H3K18R construct are all strong additions to the manuscript. One minor suggestion will be to improve the example figures for the DNA fiber assays (Fig4 and FigS2). As shown, it is not clear they reflect the major differences indicated in the quantification graphs. Overall, I believe the study is greatly improved and suitable for publication. It will be of broad interest to the EMBO readership.

Author Response to 2nd Editorial Decision

26 April 2016

We agree with your proposed modifications to the manuscript title and abstract. Thank you very much for your input.

With respect images for the DNA fiber analyses, we consider that Figure 4H perfectly reflects the reduction of fiber length that correlates with fork velocity as quantified in Figure 4I. Expanded view Figure EV2C, it is just an example of the types of labelling that can be observed upon hydroxyurea treatment as explained in the figure legend. As you acknowledged, this is just an example. In this case the results are a quantification of classifications, which is quite difficult to represent within a single image. Therefore, we are satisfied with the current version of this figure.

Acceptance

27 April 2016

Thank you for submitting your final revised manuscript for our consideration. I am pleased to inform you that we have now accepted it for publication in The EMBO Journal.

Corresponding Author Name: Lourdes Serrano

Journal Submitted to: EMBO

Manuscript Number: EMBOJ-2015-93499R